# Single-cell RNA sequencing shows the immunosuppressive landscape and tumor heterogeneity of HBV-associated hepatocellular carcinoma

Daniel Wai-Hung Ho [1,2✉], Yu-Man Tsui[1,2], Lo-Kong Chan[1,2], Karen Man-Fong Sze[1,2], Xin Zhang[1,2], Jacinth Wing-Sum Cheu[1], Yung-Tuen Chiu[1,2], Joyce Man-Fong Lee[1,2], Albert Chi-Yan Chan[2,3], Elaine Tin-Yan Cheung[4], Derek Tsz-Wai Yau[4], Nam-Hung Chia[5], Irene Lai-Oi Lo[5], Pak-Chung Sham [6], Tan-To Cheung[2,3], Carmen Chak-Lui Wong [1,2] & Irene Oi-Lin Ng [1,2✉]

Interaction between tumor cells and immune cells in the tumor microenvironment is important in cancer development. Immune cells interact with the tumor cells to shape this process. Here, we use single-cell RNA sequencing analysis to delineate the immune landscape and tumor heterogeneity in a cohort of patients with HBV-associated human hepatocellular carcinoma (HCC). We found that tumor-associated macrophages suppress tumor T cell infiltration and TIGIT-NECTIN2 interaction regulates the immunosuppressive environment. The cell state transition of immune cells towards a more immunosuppressive and exhaustive status exemplifies the overall cancer-promoting immunocellular landscape. Furthermore, the heterogeneity of global molecular profiles reveals co-existence of intra-tumoral and inter-tumoral heterogeneity, but is more apparent in the latter. This analysis of the immunosuppressive landscape and intercellular interactions provides mechanistic information for the design of efficacious immune-oncology treatments in hepatocellular carcinoma.

[1] Department of Pathology, The University of Hong Kong, Hong Kong, China. [2] State Key Laboratory of Liver Research, The University of Hong Kong, Hong Kong, China. [3] Department of Surgery, The University of Hong Kong, Hong Kong, China. [4] Department of Pathology, Queen Elizabeth Hospital, Hong Kong, China. [5] Department of Surgery, Queen Elizabeth Hospital, Hong Kong, China. [6] Department of Psychiatry, The University of Hong Kong, Hong Kong, China. ✉email: dwhho@hku.hk; iolng@hku.hk

Various oncogenic genomic, transcriptomic, and epigenetic alterations accumulate in hepatocytes through a stepwise manner, affecting various signaling pathways to drive hepatocellular carcinoma (HCC). Importantly, HCC tumor is composed of a complex tumor microenvironment (TME), consisting of cellular (tumor-infiltrating immune cells and stromal cells), chemical (chemokines), and physical components (extracellular matrix)[1]. These interact with one another to support HCC development and influence treatment responsiveness[2,3].

The fundamental understanding of the subtle cellular and molecular landscapes in HCC remains elusive. Traditional strategies perform investigation mainly at the bulk-tumor level and have inherent limitations in providing precise information on individual cells residing in a highly admixed TME. Single-cell sequencing provides an important platform to study cancers[4]. Regarding HCC, reports using single-cell analyses are few. These few reports have provided important insights, yet have limitations in giving a holistic overview to reveal the *bona fide* multifaceted landscapes and interactome. For instance, they carried out investigations in restricted immune cell types[5–7]. The study by Zheng et al. examined only the cancer stem cell (CSC) subpopulation in cell lines and one HCC case[8]. Another study focused only on the combined hepatocellular and cholangiocarcinoma (c-HCC-CC), which is an uncommon subtype of primary liver cancer[9]. One study provided a human liver cell atlas but focused primarily on the normal liver, with only limited cells sequenced on very few HCC patients[10]. The study by Ma et al.[11] was elegantly performed and studied a cohort of nine HCC cases with etiology mostly due to hepatitis C viral infection; however, the number of cells studied per case was low (~500 cells per case on average). We have recently performed our proof-of-concept exploration on the patient-derived tumor xenograft HCC model and identified stemness-related tumor cell subpopulations[12]. Due to the limitation of immune-compromised background and nonhuman nature of that model, we performed single-cell RNA sequencing (scRNA-seq) investigation on a cohort of HBV-associated HCC clinical samples to better recapitulate the actual biological processes occurring in human HCC.

In this work, we take advantage of the unique multidimensional capacity of scRNA-seq to delineate the multifaceted landscapes and cell-cell interactions in human HCCs. We show the cellular and immunosuppressive landscapes that suggest functions of tumor-associated macrophages and TIGIT–NECTIN2 interaction in shaping a cancer-promoting immunosuppressive environment in hepatocarcinogenesis. These findings specifically indicate molecular targets for subsequent translational applications. The inter- and intra-tumor heterogeneity and the non-negligible level of intra-tumoral heterogeneity may provide evidence and insight in supporting the adoption of immune checkpoint inhibitors (ICIs) over or in combination with receptor tyrosine kinase (RTK) inhibitors for treating advanced HCC. Our analysis of the immunosuppressive landscape and intercellular interactions provide useful mechanistic information for the design of efficacious immune-oncology treatments in HCC.

## Results

**scRNA-sequencing of clinical HBV-associated HCC patients.** The demographic and pathological data of the eight randomly selected HCC cases are shown in Supplementary Table 1. We performed the scRNA-seq experiments using the Chromium platform (10× Genomics) to isolate the single cells from HCC samples (Fig. 1a). This platform relies on microfluidics circuits to capture single cells. We estimated the cell multiplet rate by mixing equal numbers of human and mouse cells and estimated it

to be only 0.6% (Supplementary Fig. 1a). Pooled and barcoded single-cells in the form of droplets were then subjected to subsequent manipulation procedures for library preparation. Sequencing libraries of pooled single cells were obtained and subjected to a sequencing run. We performed the initial analysis to obtain the estimated cell count for each sample (Supplementary Table 2). We applied additional quality control filtering, including minimum gene count, maximum mitochondrial percentage, and multiplet removal, on the data (Supplementary Fig. 1b). The resulting cell counts are summarized in Supplementary Table 2. We further performed normalization and scaling, followed by batch effect checking (Supplementary Fig. 2) to ensure there were no significant technical differences between samples.

**Global cellular and transcriptomic landscapes.** With the cleaned dataset on the eight clinical HCC cases, we conducted principal component analysis (PCA) using the most highly variable genes ($n = 5201$) (Supplementary Fig. 3a) and computed the linear dimension reduction on the first 100 principal components (PCs). Taking the PC heatmap (Supplementary Fig. 3b) and PC elbow plot (Supplementary Fig. 3c) into consideration, we proceeded to include the first 30 PCs for the subsequent analyses. To visualize these data, we applied a uniform manifold approximation and projection (UMAP) algorithm[13] to perform nonlinear dimensionality reduction. The single cells were stratified according to their global transcriptomic landscape in two-dimension space, indicating their gene expressional similarity/difference (Fig. 1b). The results indicated some cell clusters contained single cells from multiple cases, while some consisted of cells primarily from an individual case. Upon examining the expression of cell-type markers, HCC tumor markers, and liver CSC (LCSC) markers (Supplementary Table 3) in the single cells, the cell type identity was identified (Supplementary Figs. 4 and 5). It was found that malignant and nonmalignant cells clustered into distinct cell clusters, with malignant cells (tumor cells) from different cases separated into different cell clusters while nonmalignant cells from multiple cases admixed together (Fig. 1b). We then clustered the single cells into 34 different cell clusters using a graph-based Louvain clustering algorithm on the k-nearest-neighbor (KNN) graph (Fig. 1c). Differential expression analysis was then performed on individual cell clusters against the remaining cells. We were able to detect specific signature genes to suggest molecular characteristics of cells from individual cell clusters, which guided subsequent cell type identification (Supplementary Fig. 6).

**Cell type heterogeneity.** We assigned the identity of different cell clusters by jointly considering the spatial stratification pattern in the UMAP plot (Fig. 1c) and the expression of different gene markers (Supplementary Fig. 7a). The tumor cells from the same case, rather than from different cases, tended to cluster together, suggesting a relatively high level of inter-tumoral heterogeneity (Fig. 1b). Different HCC cases had different proportions of immune cells (Fig. 1d and Supplementary Table 4).

**The inverse relationship between tumor-infiltrating T cells and tumor-associated macrophages.** We observed apparent contrasting and possibly inversely correlated proportions of T cells and macrophages in these cases. Indeed, the proportion of T cells (and the subset of CD8 T cells) and macrophages (and the subset of CD163 M2 macrophages) were extracted, we detected a suggestive inverse correlation (Fig. 1e). To verify this intriguing relationship between tumor-infiltrating T cells and macrophages, we performed immune cell deconvolution[14] on the bulk-cell RNA-seq data from our in-house HBV-associated HCC

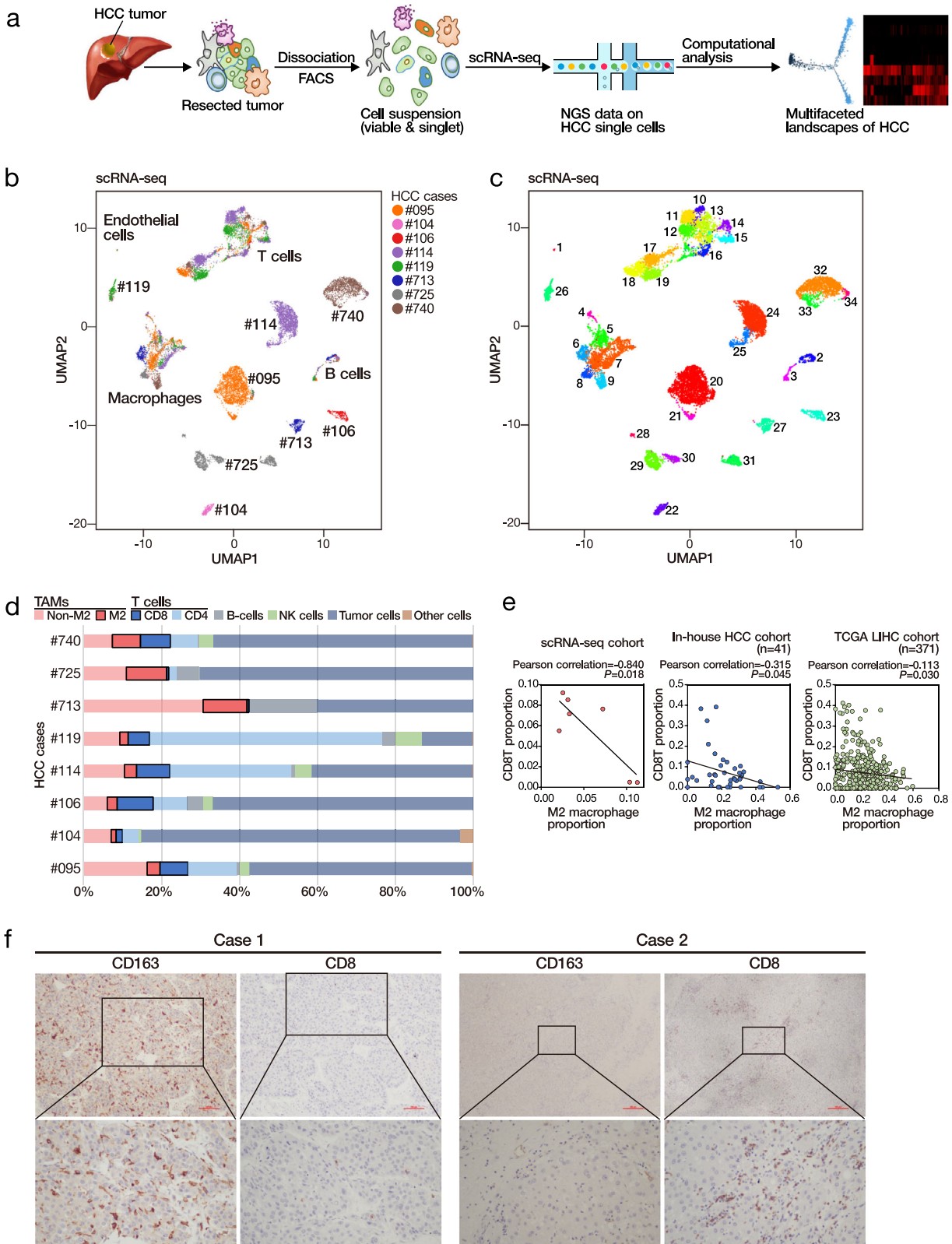

**Fig. 1 scRNA-seq suggests inverse correlation between TAM and CD8 T cells. a** Workflow of the experiment. **b** Stratification and cell-type identification of HCC single cells. Malignant cells (labeled by the case identity) and non-malignant cells (labeled by cell types) were grouped into distinctive cell clusters. **c** Single cells from the 8 HCC cases were stratified into 34 cell clusters using a graph-based Louvain clustering algorithm on the KNN graph (resolution = 1.2). **d** Summary of different major cell types identified in HCC tumors. **e** Proportion of tumor-infiltrating macrophages was inversely correlated to the proportion of tumor-infiltrating T cells in scRNA-seq and deconvoluted bulk-cell RNA-seq datasets. Pearson correlation (two-sided). **f** Immunohistochemistry of CD8 T cells and CD163 M2 macrophages in two representative HCC cases of an independent cohort. Scale bar = 100 μm. More than ten fields each under ×40 and ×100 magnification were examined. Source data are provided as a Source Data file.

database[12,15] as well as TCGA LIHC database. We were able to observe a consistent trend as in our initial observation on this inverse correlation between the proportions of tumor-infiltrating T cells and macrophages, suggesting the possibility that the abundance of tumor-associated macrophages (TAMs) might predict the scarcity of tumor-infiltrating T cells (Fig. 1e, f and Supplementary Fig. 8).

**Immunosuppressive marker expression of TAMs and the prognostic significance.** We examined the expression of a panel of reported immunosuppressive molecules in TAMs and found the enriched and likely restricted expression of *LAIR1*, *HAVCR2* (also known as *TIM3*), *LGALS9*, and *VSIR* in TAMs (Fig. 2a and Supplementary Fig. 7b). Given that the enrichment pattern of *CD163* (M2 macrophage marker) coincided nicely with that of *LAIR1* in different macrophage cell clusters (Fig. 2b), we postulated the immunosuppressive function of TAMs might be exerted via *LAIR1*. To this end, we also detected a statistically significant association between *CD163* and *LAIR1* expressions in both our in-house and TCGA RNA-seq datasets, suggesting a putative immunosuppressive role of *LAIR1* in M2 macrophages (Fig. 2c). In addition, further investigation using the TCGA LIHC cohort (via cBioPortal with the default *z*-score expression threshold of 2) showed that the high expression of *LAIR1* and *HAVCR2* each was significantly associated with poorer overall or disease-free survival of HCC patients (Fig. 2d). We also observed frequent overlap of the LAIR1 and M2 macrophages (CD163) using multicolor immunofluorescence staining (Fig. 2e). Consistently, with immunohistochemistry (IHC) on a cohort of HCC (*n* = 29), we also detected a statistically significant association between LAIR1 and CD163 expression in HCC (Supplementary Table 5). There was also frequent co-expression of LAIR1 and CD163 (Supplementary Fig. 9). All these findings support our hypothesis of enriched LAIR1 expression in M2 macrophages. Moreover, we established stable *LAIR1* KD (sh*LAIR1*) macrophages using THP-1 cells. Upon sh*LAIR1*, THP-1 cells demonstrated reduced proliferation. By coculturing them with CD8 T cells, we also identified upregulated T cell activation, as exemplified by the increased proportion of CD44+ CD62L− effector T cells (Supplementary Fig. 10).

**Cell-state transition trajectory of different tumor-infiltrating immune cell subpopulations.** To understand the underlying evolvement of cellular status in the immune TME, we derived the pseudo-time cell trajectory of the various tumor-infiltrating immune cells, including CD8 T, CD4 T, regulatory T cells (Treg), natural killer (NK), and TAMs (Fig. 3). Interestingly, by combining the findings from both the clustering and pseudo-time analyses, we observed the gradual transition of CD8 T and NK cells towards the subpopulations with exhaustion status. Such exhaustion status for CD8 T cells was indicated by the upregulation of *PDCD1* (*PD-1*) and *TIGIT* (T cell immunoreceptor with Ig and ITIM domains) in cell cluster 16 and for NK cells by downregulation of *FCGR3A* (activation marker) and upregulation of *KLRC1* (exhaustion marker) in cell cluster 14. For Treg cells, they also showed the transition toward a more immunosuppressive state, as indicated by the enriched *PDCD1* expression in cell cluster 17. Coinciding with our previous observation on TAMs, these findings also indicate a dynamic transition towards more immunosuppressive M2 macrophages featured by *CD163* and *LAIR1* expression in cell clusters 6–9. Regarding the CD4 T cells, they were mainly type 1 (Th1) and type 2T helper (Th2) cells, as indicated by their respective enrichment in the expression of *STAT4* (cell cluster 11) and *GATA3* (cell cluster 12). Taken together, we identified the dynamics in the cell state transition of

the immune cells towards a more exhaustive or immunosuppressive status with distinctive gene expression signatures. This exemplified the overall immunosuppressive cellular landscape in HCC tumors supportive of evading immune surveillance and HCC development.

**The immunosuppressive landscape of immune checkpoints.** Immune checkpoints function as "brakes" on T cell immune responses resulting in the weakening of T cell attack and immune tolerance/escape. We examined the expression of immune checkpoint components in both tumor-infiltrating immune cells, i.e., CD4 T, CD8 T, NK, and Treg cells, and the complementary antigen-presenting cells (APCs), i.e., macrophages and tumor cells. Upon evaluating the cell–cell interaction status by jointly considering the gene expression pattern of immune checkpoint components in the complementary cells, e.g., *PDCD1* (*PD-1*) in T cells, and *CD274* (*PDL1*) and *PDCD1LG2* (*PDL2*) in the tumor cells, we estimated the relative contribution of different co-stimulatory and co-inhibitory checkpoints in shaping the immunosuppressive landscape in HCC (Fig. 4a and Supplementary Fig. 11). In general, the estimated levels of interaction via co-inhibitory checkpoints were higher than those of co-stimulatory checkpoints. In particular, we identified a prominent co-inhibitory signal via the *TIGIT–NECTIN2* (Nectin Cell Adhesion Molecule 2) axis in complementary T cells and APCs (Fig. 4a, b). Besides, *NECTIN2* (also called *PVRL2*) was the most highly expressed gene in the *PVR* gene family in these cases (Supplementary Fig. 12). Furthermore, in both our in-house and TCGA cohorts, NECTIN2 was significantly upregulated in HCCs as compared to the corresponding non-tumorous livers (Fig. 4c). Furthermore, we performed IHC for TIGIT and NECTIN2 expression on a cohort of HCC (*n* = 29) and a cohort of non-HCC, HBV-associated cirrhotic liver (*n* = 22). We examined the expression of TIGIT and NECTIN2 in the lymphocytes and tumor cells, respectively (Supplementary Figure 13). Interestingly, we detected a statistically significant association between TIGIT and NECTIN2 expression in HCC but not in non-HCC, HBV-associated cirrhotic livers (Supplementary Tables 6 and 7). This suggests that the TIGIT-NECTIN2 axis may likely be a tumor evasion strategy, instead of viral evasion one. Moreover, using IHC, we demonstrated the significant upregulation of NECTIN2 in HCCs, as compared to the corresponding non-tumorous livers (Supplementary Fig. 14). These findings suggest that overexpression of NECTIN2 might be important in building up the immunosuppressive landscape in HCC development (see below for Experimental results on NECTIN2 and TIGIT). We also detected substantial expression of multiple immunosuppressive immune checkpoint genes (*HAVCR2*, *CTLA4*, and *TIGIT*) in Treg cells (Fig. 4a). Moreover, while the interaction between *PD-1* (*PDCD1*) and *PD-L1/L2* (*CD274*/*PDCD1LG2*) was minimal, the co-inhibitory signal via *CTLA4-CD86* and *HAVCR2-LGALS9* axes in Treg cells and macrophages, respectively, was substantial (Supplementary Fig. 11).

**TIGIT–NECTIN2 axis in HCC.** Upon ligation to PVR/NECTIN2, TIGIT acts as an inhibitory receptor on T and NK cells. To investigate whether NECTIN2 expressed on HCC cells would lead to T cell exhaustion, we established *Nectin2* knockout (KO) stable cells in a mouse HCC cell line, Hepa1–6, with the CRISPR–Cas9 KO system using two independent single-guide RNAs (sgRNAs). We isolated mouse splenic T cells and cocultured with or without Hepa1–6 parental cells in the presence or absence of anti-Nectin2 neutralizing antibody (anti-Nectin2 ab). The presence of mouse HCC cells in the coculturing system significantly inhibited the proliferation of CD4+ or CD8+ T cells as compared to culturing

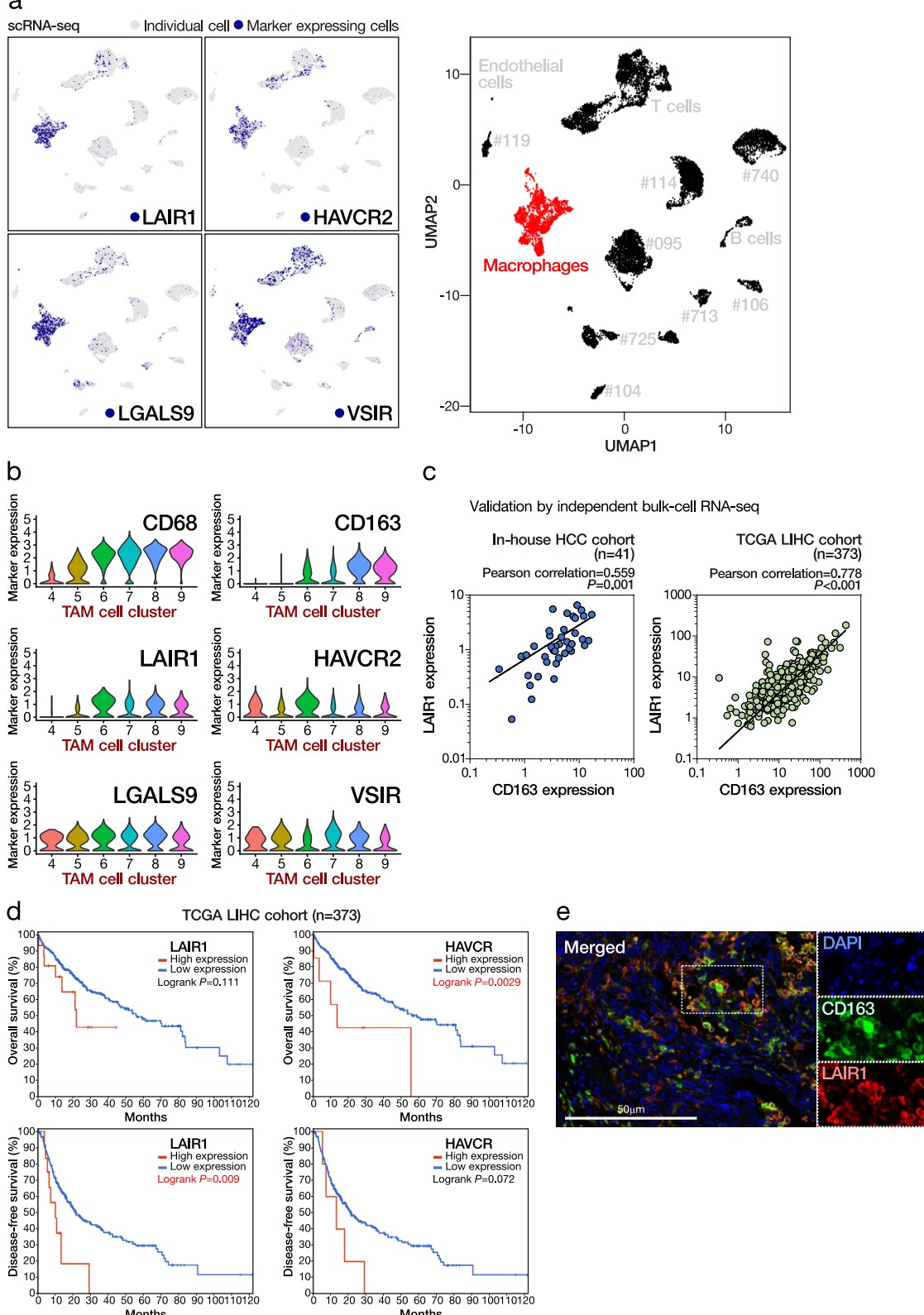

**Fig. 2 High expression of TAM markers in HCC tumors suggests poor prognosis. a** TAMs had enriched expression for multiple immunosuppressive markers. **b** TAM cell clusters expressing cancer-promoting M2 macrophage marker *CD163* had concurrent enrichment for *LAIR1* expression. **c** Expressions of *CD163* and *LAIR1* were significantly correlated in both in-house and TCGA datasets. Student's *t* test (two-sided). **d** High expressions of *LAIR1* and *HAVCR2* were significant associated with poorer disease-free and overall survivals, respectively. Log-rank test (two-sided). **e** Overlap of the LAIR1 and M2 macrophages (CD163) using multicolor immunofluorescence staining in Case #713. Scale bar = 50 μm. More than ten fields each under ×40 and ×100 magnification were examined. Source data are provided as a Source Data file.

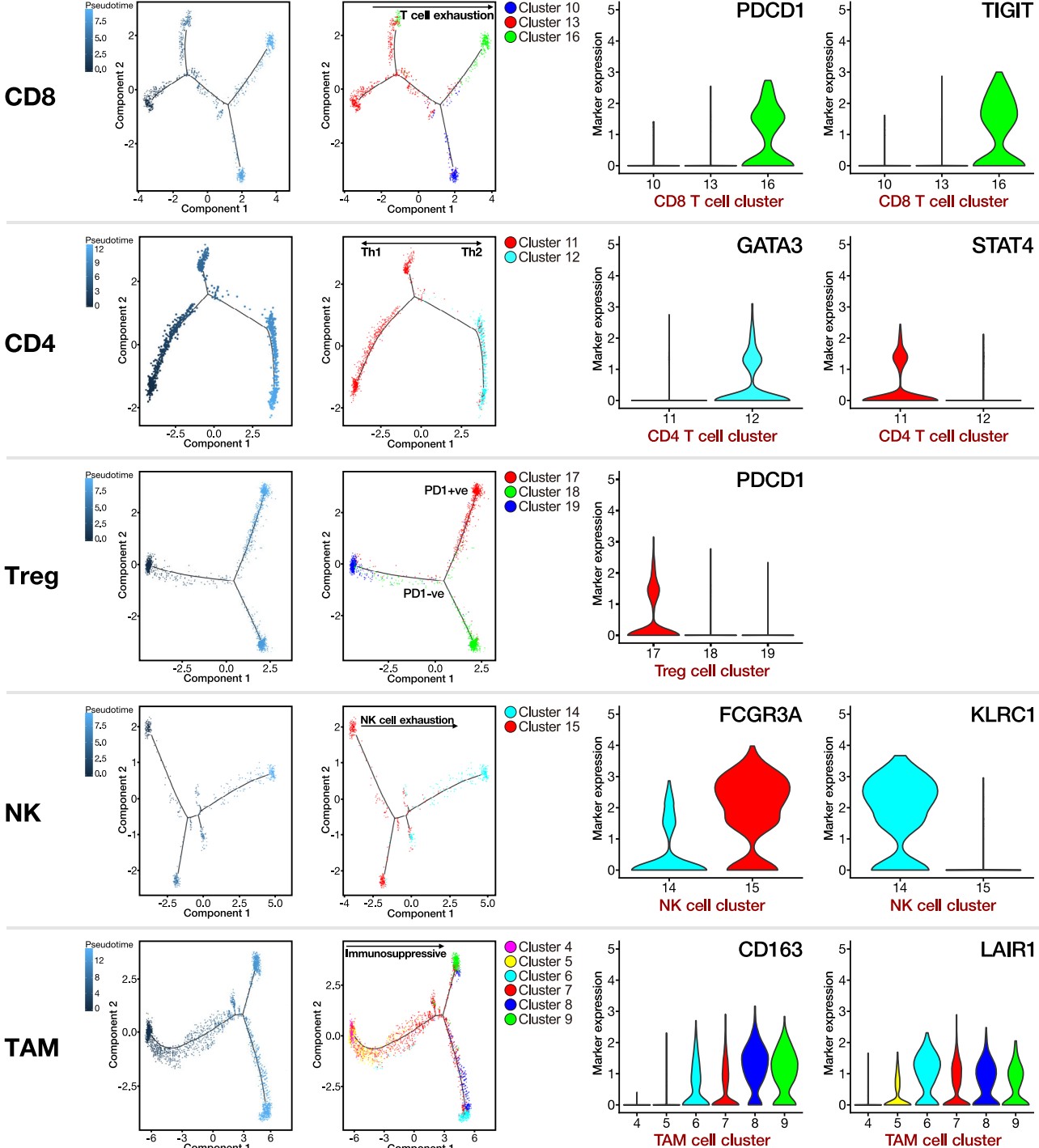

**Fig. 3 Pseudotime cell trajectory analysis on major immune cell types.** We correlated the cell trajectory analysis with the clustering result on various immune cell types. Representative markers indicate their overall transition to more immunosuppressive and exhausted status.

T cells alone (Fig. 5a). Anti-Nectin2 ab significantly restored both CD4+ and CD8+ T cell proliferation in the cocultuing system (Fig. 5a). We further cocultured mouse splenic T cells with Hepa1–6–WT, Hepa1–6-Nectin2–KO-1 (KO1), Hepa1–6-Nectin2–KO-2 (KO2), or Hepa1–6–Nectin2–KO-3 (KO3) stable HCC cells. Consistently, KO of Nectin2 restored both CD4 and CD8 T cell proliferation, suggesting that Nectin2 suppressed T cell proliferation (Fig. 5b). To extend our observation in vivo, we performed hydrodynamic tail–vein injection (HDTVi) to gen-erate Nectin2 WT and Nectin2 KO mouse HCC with the genetic background of Tp53 KO and c-Myc overexpression, as we

previously described[16]. Five weeks after HDTVi, we found that the Nectin2 KO HCC tumors were significantly smaller in size as compared to Nectin2 WT HCC tumors (Fig. 5c). Tumor shrinkage was accompanied by increased infiltration of T effector, CD4+, and CD8+ cells in HCC, as confirmed by flow cytometry of the tumor-infiltrated lymphocytes (TILs) (Fig. 5d–f and Sup-plementary Fig. 15a). IHC staining consistently confirmed the increase of CD4+ and CD8+ T cells in the Nectin2 KO HCC tissues as compared to Nectin2 WT HCC tissues (Fig. 5g, h). Expression of exhaustion markers including PD-1, TIGIT, LAG3, and TIM3 on T cells did not significantly alter (Supplementary

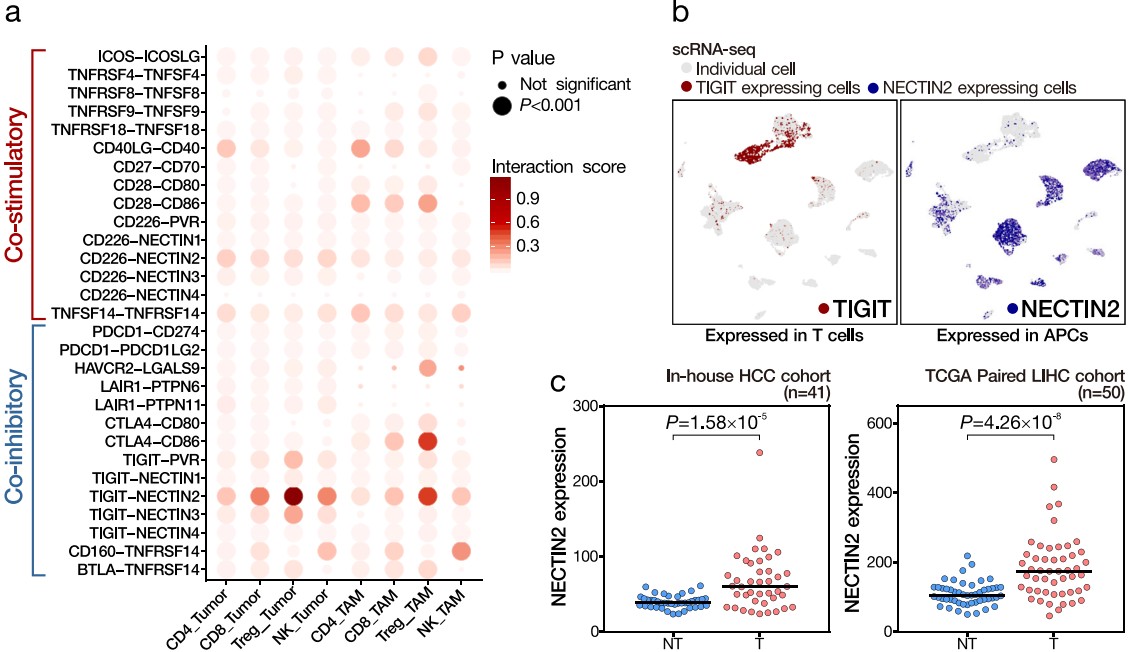

**Fig. 4 Immune checkpoint analysis in HCC implicates *TIGIT–NECTIN2* interaction. a** We examined the immune checkpoint interactions between lymphocytes and APCs (tumor cells and TAMs) and identified the prominent interaction via the *TIGIT–NECTIN2* axis (circle size indicates the statistical significance and circle color indicates the level of interaction). The empirical *P* value was estimated by 1000 imputations. **b** The expression of *TIGIT* and *NECTIN2* was respectively enriched in T cells and APCs. **c** Upregulation of *NECTIN2* was detected in HCC tumors, as compared to non-tumorous livers in both in-house and TCGA datasets. Student's *t* test (2-sided). Source data are provided as a Source Data file.

Fig. 15b). To confirm our observation in another mouse HCC model, we orthotopically implanted Hepa1–6–NTC and Hepa1–6–sh*Nectin2* mouse HCC cells into syngeneic immune-competent mice. Consistent with KO of *Nectin2*, knockdown (KD) of *Nectin2* suppressed HCC growth and restored CD4$^+$ and CD8$^+$ T cell infiltration (Supplementary Fig. 15c–e). These data together highlighted the function of *NECTIN2* in limiting T cell infiltration and activity in HCC.

**Cell–cell interactome landscape via ligand–receptor interactions.** Different cell types exist in the immune TME of HCC. To examine and quantify the ligand–receptor interaction between different cell types, we implemented the method proposed by Kumar et al.[17] to calculate an interaction score that evaluated the degree of cell–cell communication according to different ligand–receptor relationships. We found a relatively low level of interaction between tumor cells and B cells (particularly when tumor cells contributed ligands and B cells contributed receptors). Remarkably, tumor cells frequently interacted with various immune cells. This phenomenon of interaction with immune cells was similarly detected in TAMs, but to a lesser extent. In addition, there was a relatively high level of interaction between tumor cells and TAMs, particularly via the MHC I-LILRB axis, where MHC class I molecules (B2M and HLA) and LILRB molecules (LILRB1 and LILRB2) were expressed in the respective cells for immunosuppression that supports tumor growth in other cancers[18,19] (Supplementary Fig. 16).

**Subclonal heterogeneity and lineage hierarchy in HCC tumor cells.** We stratified the HCC tumor cells according to their global transcriptomic profile. The results showed that HCC tumor cells mainly grouped together according to their case identity, with very few cells admixed into clusters from other cases (Supplementary Fig. 17). This indicates HCC tumor cells from the same

patient shared a higher degree of similarity than those from different patients. Furthermore, we defined them into different LCSC marker groups, according to the unsupervised hierarchical clustering of gene expression pattern for a panel of LCSC markers (*EPCAM*, *KRT19*, *ALDH1A1*, *CD13*, *CD24*, *CD44*, *CD47*, *CD90*, and *CD133*) (Fig. 6a). The distribution of the seven groups of HCC tumor cells (LCSC marker groups 1–7) in the eight HCC cases is summarized in Supplementary Table 8. Each case was detected to have one or two major LCSC marker groups of HCC tumor cells (Supplementary Fig. 18). Interestingly, although there was a modest correlation between case identity and LCSC marker group status, we observed that the LCSC marker group 2 cells, which concurrently expressed *EPCAM*, *ALDH1A1*, and *CD24*, grouped together into cell cluster and contained HCC tumor cells from multiple cases. They might represent a molecularly distinct tumor cell subpopulation and this awaits further experimental investigation.

To further explore the heterogeneity landscape of HCC tumor cells, we also inferred copy-number variation (CNV) status based on the modified algorithm of Tirosh et al.[20] We generated the genome-wide CNV alteration map for the HCC tumor cells from each case and derived the lineage hierarchy of HCC tumor cells based on inferred CNV alterations (Supplementary Fig. 19). Similar to LCSC marker group identification, the genome-wide CNV profile was used to classify the HCC tumor cells into different CNV groups (Supplementary Fig. 20). The results showed that there were 9 major groups of HCC tumor cells (CNV groups 1–9) based on the inferred CNV status (Fig. 6b), and each HCC case was enriched with one major CNV group of tumor cells (except Case #725 which had 2 major CNV groups of tumor cells) (Supplementary Table 9). We further examined the expression profile of RTK gene families in HCC tumor cells (Supplementary Fig. 21). RTK gene expression was enriched in a patient-specific manner. Overall, these findings collectively suggested that the degree of inter-tumoral heterogeneity was

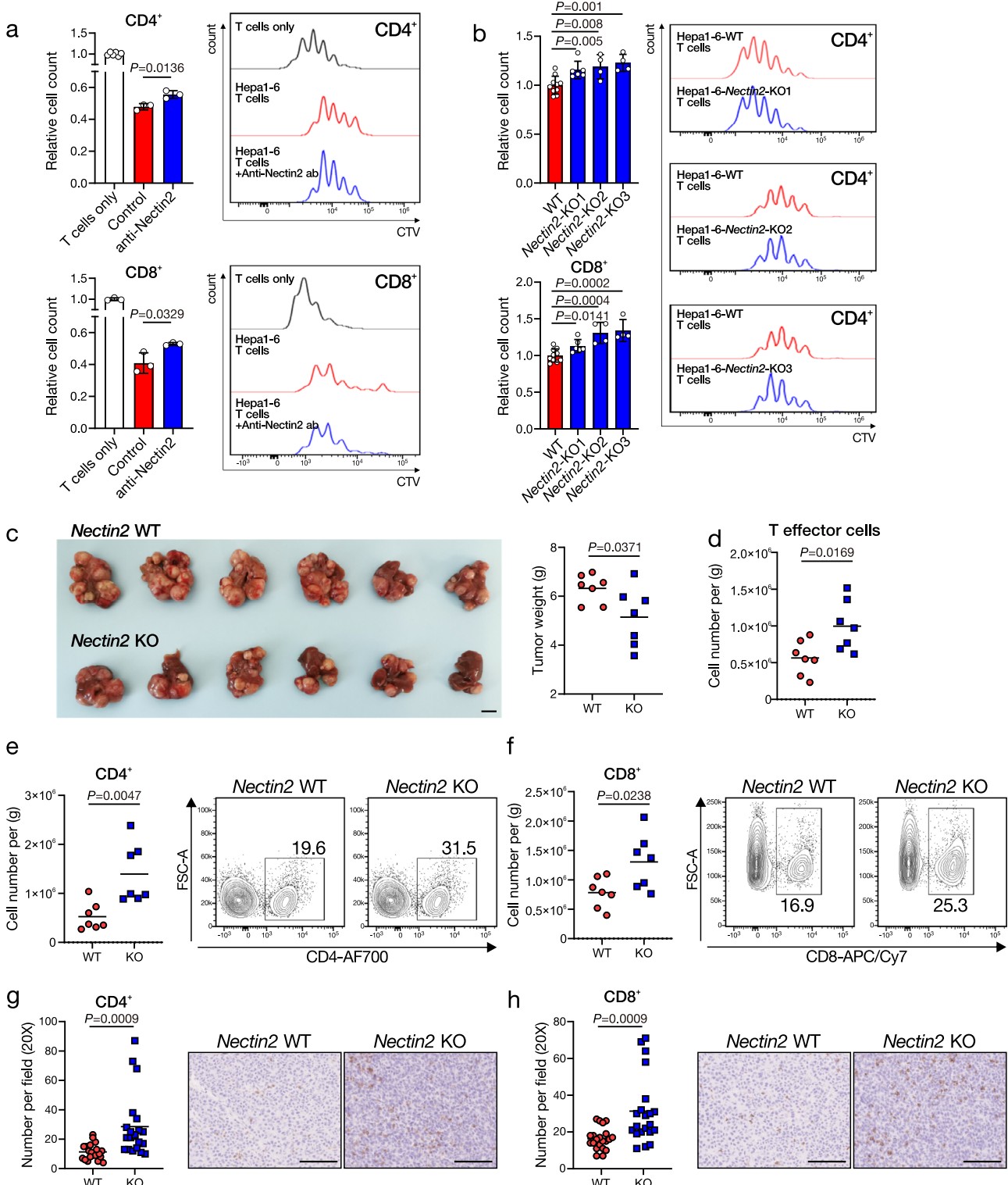

**Fig. 5 NECTIN2 in HCC cells suppresses T cell activity. a** CellTrace Violet (CTV)-labeled mouse splenic T cells were isolated and co-cultured with Hepa1–6 cells in the presence or absence of anti-Nectin2 neutralizing antibody (15 µg/mL). Mean ± SD is presented. **b** CTV-labeled T cells were cocultured with Hepa1–6 (WT), -Nectin2-KO1, -Nectin2-KO2, -Nectin2-KO3 cells. Mean ± SD is presented. **c** Representative picture and weight of Nectin2 WT (Nectin2WT:Tp53KO:c-MycOE), and Nectin2 KO (Nectin2KO:Tp53KO:c-MycOE) HCC tumors. Scale bar = 1 cm. **d–f** Numbers of tumor-infiltrating lymphocytes were analyzed by flow cytometry. **g, h** Representative pictures, and quantification of CD4 + T cells and CD8 + T cells in HCC tumors by IHC staining. Scale bar = 100µm in IHC representative pictures. **a–h** Student's t test. The experiment was performed with a variable number of biologically independent samples (n number) (**a** n = 6, n = 3, and n = 3 for T cells only Ctrl and anti-Nectin2 respectively in CD4+ cells, and n = 3 for all groups in CD8+ cells; **b** n = 10, n = 6, n = 4, and n = 4 for WT, Nectin2-KO1, Nectin2-KO2, and Nectin2-KO3, respectively in both CD4+ and CD8+ cells; **c–e**, **g**: n = 7 per group; **f, h**: n = 21 per group). Source data are provided as a Source Data file.

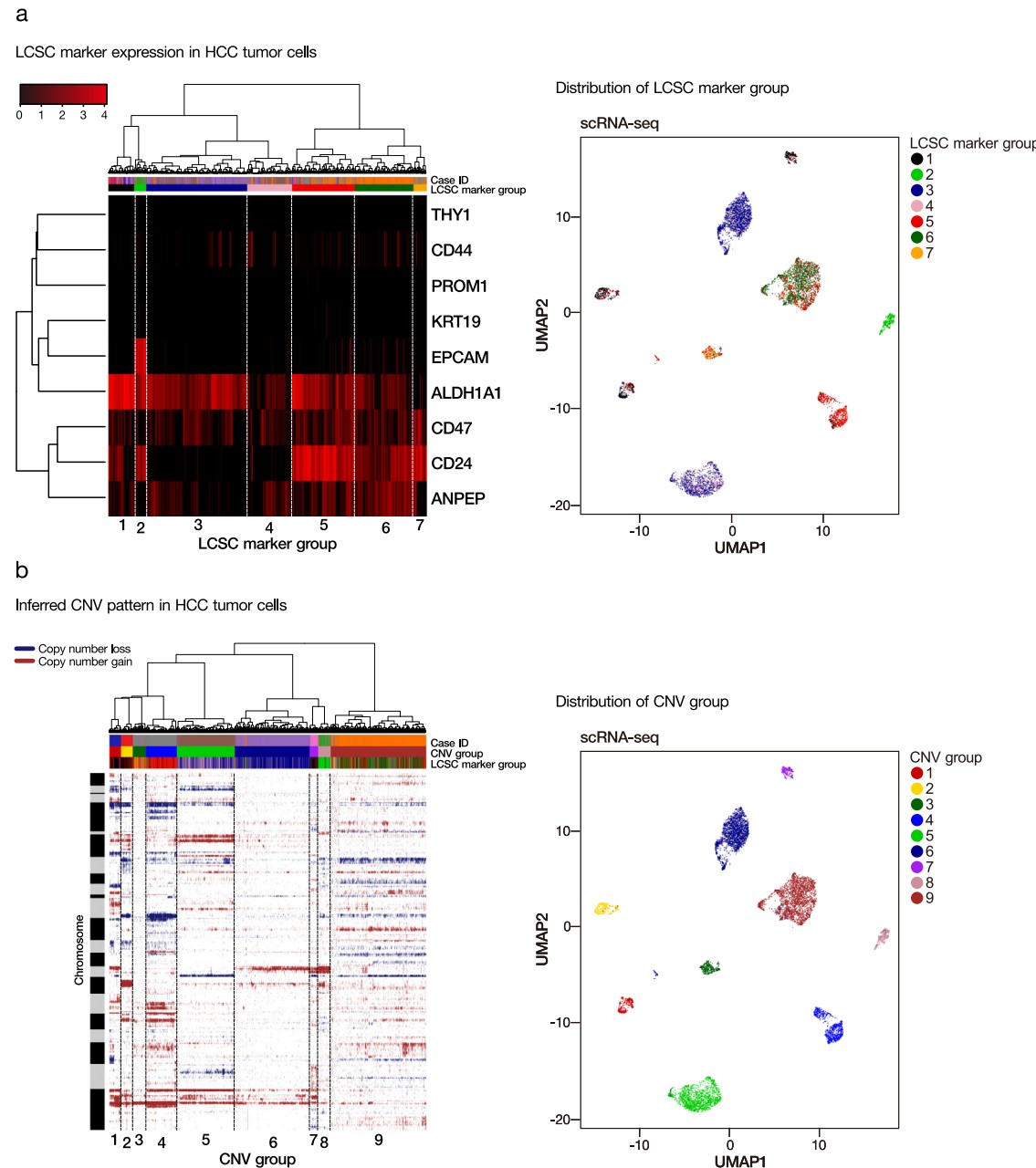

**Fig. 6 Tumor heterogeneity in terms of LCSC marker and inferred CNV status. a** We stratified the HCC tumor cells according to LCSC marker expression into different LCSC marker groups (left). The distribution of LCSC marker groups was displayed in the UMAP plot (right). **b** HCC tumor cells were also stratified according to inferred CNV patterns into different CNV groups (left). The distribution of CNV groups was displayed in the UMAP plot (right).

more prominent than the intra-tumoral one, since HCC tumor cells from individual cases tended to cluster together according to similarities in terms of global transcriptomic profile, LCSC markers, inferred CNV status, and RTK expression. However, there were major and minor cell subclones in each HCC tumor, indicting a non-negligible level of intra-tumoral heterogeneity that could impose difficulty in molecularly targeted therapy that usually pinpoints the major subclone.

## Discussion

Currently available systemic treatments for advanced HCC target two different vulnerabilities in HCC, which include RTK inhibitors (sorafenib, regorafenib, lenvatinib, cabozantinib, and ramu-cirumab) and ICIs (nivolumab and pembrolizumab, both being

anti-PD1 antibodies)[21]. However, their efficacy is far from adequate, and the objective response rates are still low (at most 25%)[22], indicating our insufficient understanding of human HCC that hinders effective drug development. Immune checkpoints are mechanisms that hamper T cell immune responses and there are ICIs that release these "brakes" to reactivate T cells to fight against cancer. ICIs offer better treatment responses for advanced HCC patients. Very recently, US FDA has granted expedited approval for atezolizumab (anti-PD-L1) in combination with bevacizumab (anti-VEGF) as first-line treatment for advanced inoperable HCC[23]. Indeed, the degree of T cell infiltration in tumors defines T cell[high]/hot tumor or T cell[low]/cold tumor[24–26] and studies have suggested differential responsiveness towards ICI treatment for hot and cold tumors[27,28]. Besides, the presence

of TILs could predict a better prognosis for HCC[29], indicating the enhancement of tumor T cell infiltration, particularly the CD8 T cells, may provide a potential benefit for HCC patients. Intriguingly, our current study has revealed that the presence of TAMs, particularly the cancer-promoting M2 TAM, could predict an inverse relation with tumor-infiltrating (CD8) T cells, indicating the potential hindrance effect imposed by TAMs on T cell infiltration. Our finding coincides with a previous report on macrophages impeding T cells from reaching tumor cells in lung cancer[30].

The intricate relationships between HCC cells and various immune cell types in fact shape the immune TME in HCC. Studies have found that HCC cells secrete osteopontin (OPN)[31], sonic hedgehog (SHH)[32], and GP73 (produced upon ER stress)[33] to promote M2 polarization of TAMs. Furthermore, CCL2/CCR2 signaling in HCC cells could promote the recruitment of M2-polarized TAMs, malignant growth, and metastasis[34]. On the other hand, M2-like TAMs secrete higher IGF-1[35], lower CXCL9 and CXCL10[32], and higher CCL20 (from TREM-1+ TAMs under hypoxia) to inhibit CD8 T cells and promote the recruitment of regulatory Treg cells to bring about immunosuppression and poorer survival[36]. Of note, in this study, we observed that not only was the expression of the key immunosuppressive targets (LAIR1 and HAVCR2) enriched in TAMs but their high expression in HCCs was associated with adverse clinical prognoses. LAIR1 (CD305) is a transmembrane glycoprotein[37,38], which is constitutively expressed on most mononuclear leukocytes[39–41] and some cancer cells[42–44]. The cross-linking of LAIR1 on immune cells poses inhibitory functions[45]. There were contradictory findings regarding the role of LAIR1 in M2 polarization[46–48] and up till now, there are no reports addressing the role of LAIR1 in TAM specifically in HCC. HAVCR2's immunosuppressive role has been well studied in infiltrating immune cells in HCC[49–52]. Its enhanced expression in CD14+ monocytes and CD206+ TAMs in HCC was associated with poorer survival[51]. Taken together, our findings provide important insight suggesting the involvement of cancer-promoting M2 TAMs in hepatocarcinogenesis via their immunosuppressive functions against T cell infiltration and activation. Agents targeting the different aspects of TAMs, ranging from macrophage depletion, recruitment, and polarization, might provide solutions in creating a favorable TME to support ICI treatment in HCC[53].

By identifying different cell types in HCC tumors, we evaluated the relative importance of the different immune checkpoint molecules in human HCC. While the PD-1-PD-L1/L2 and CTLA4-CD80/86 axes are regarded as key immune checkpoint components in various cancers, our findings revealed that they were lowly expressed and this might raise an uncertainty of their functions in treatment-naïve HCC patients. This could partially explain the incomplete efficacy of nivolumab and pembrolizumab (both anti-PD-1 antibodies), or ipilimumab and tremelimumab (both anti-CTLA4 antibodies)[54–56]. Additional immune checkpoint markers are needed for the prediction of treatment response. Notably, our findings demonstrated the prominent involvement of the TIGIT–NECTIN2 axis in HBV-associated HCC. These echo the recent findings derived primarily from mouse models by Zong et al.[57] that TIGIT is critical in adaptive immunotolerance to HBV and Ostroumov et al.[58] that TIGIT represents an important T cell exhaustion marker in liver cancer and its expression can more reliably identify exhausted CD8 T cells as compared to PD-1. In fact, we have recently also reported the significant functions of the TIGIT–PVR/PVRL1 (aka NEC-TIN1) axis and the potential therapeutic effect upon its blockade in HCC[16]. More importantly, in our current study, we

have demonstrated that TIGIT–NECTIN2 axis is a major co-inhibitory immune checkpoint in human HCC. Nectin2 KO and KD HCC mouse models consistently elicited tumors of reduced size, restored T cell infiltration, and alleviated T cell exhaustion, further exemplifying the therapeutic potential of targeting TIGIT–NECTIN2 axis, either as monotherapy or in combination with other drugs to treat advanced HCC. In fact, there are ongoing clinical trials with anti-TIGIT antibodies. For instance, the Phase II CITYSCAPE trial evaluated tiragolumab (anti-TIGIT antibody) in combination with atezolizumab (anti-PD-L1 antibody) in nonsmall cell lung cancer showed encouraging treatment outcomes[59]. It is highly anticipated that anti-TIGIT immunotherapy may offer hope to HCC patients.

Apart from revealing the inverse relationship between tumor-filtrating T cells and TAMs, our data also suggested the possible transition and accumulation of immunosuppressive and exhausted landscapes on a wide spectrum of TILs. In particular, TAMs might lead to a T cell-excluded environment by preventing CD8 T cell infiltration while Treg cells might further render the tumor-infiltrating T cells and NK cells to become exhausted from delivering their native tumor-suppressive functions. Moreover, the high degree of cell–cell interactions between tumor cells and various immune cells shapes a favorable environment for HCC development, e.g., the MHC I-LILRB axis between tumor cells and TAMs[19]. Our scRNA-seq data have provided useful cell atlas and resources to reveal the high-dimensional ecosystem in human HCC.

Focusing on the tumor cells in this study, it is worth noting that there was a significant level of inter-tumoral heterogeneity. Hence, patient stratification, ideally by specific molecular biomarkers, is pivotal to identify the most susceptible subset of HCC patients for relevant precision treatment, as contrasted to conventional chemotherapy which operates on a one-size-fits-all manner and is proven ineffective in HCC. Nonetheless, our findings also indicate there is still a non-negligible degree of intratumoral heterogeneity[60] that makes molecular targeted drugs, e.g., RTK inhibitors, eventually acquire resistance through a clonal selection of residual subclones of tumor cells. Taken together, by collectively considering the heterogeneity landscapes in terms of global transcriptomic profiles (LCSC marker panel, inferred CNV status, and RTK expression), we found the co-existence of intra- and inter-tumoral heterogeneity, with the latter being far more profound. The presence of rare subclones of tumor cells (though minor however non-negligible) may deem the eventual failure of RTK inhibitors. In this sense, ICIs or other precision treatments targeting not entirely on certain specific subclones of tumor cells will offer a more rational option for advanced HCC patients.

In summary, we report our scRNA-seq findings in the delineation of the underlying cellular composition, subclonal diversity, and high-resolution multifaceted landscapes of individual cells and, in particular, the immunosuppressive landscape and tumor heterogeneity of human HCC.

## Methods

**Tissue samples from patients with HCC**. Eight HBV-associated HCC cases surgically resected from patients were randomly selected with the following criteria: (1) HBV-associated, and (2) more than 1000 viable single cells obtained in the dissociated tumor cell suspension in each case. HBV-associated HCCs were selected to have a uniform etiological background of HCC because >80% of our HCC cases locally are associated with chronic HBV infection. The HCC tumor tissues were obtained immediately in the operation theater after surgical resection at Queen Mary and Queen Elizabeth Hospitals of Hong Kong. The study was approved by the Institutional Review Board of the University of Hong Kong/Hospital Authority Hong Kong West Cluster (UW 17-056) and informed consent was obtained from patients.

**Dissociation of HCC tumors and selection of viable singlet cells**. We followed our established procedures to dissociate the HCC tumor tissue[12]. Briefly, tumor tissue was cut into smaller pieces and further minced in medium with the addition of 10–16 μM ROCK inhibitor Y-27632 (S1049, Selleckchem, Houston, USA). The content was transferred to gentleMACS C tube (130-093-237, Miltenyi Biotec, Germany), and DNase (11284932001, Roche, Switzerland) and liberase (5401119001, Roche, Switzerland) were added. Dissociation was carried out by the automated gentleMACS dissociator (130-093-235, Miltenyi Biotec, Germany) using the built-in program specific for dissociating human tumor tissues. The C tube was incubated at 37 °C for 5–15 min in between steps of the dissociation program. The content was then filtered through a 100 μm-pore size cell strainer. The cells were spun down by centrifugation at 100rcf for 4 min and the supernatant was discarded. These steps were repeated two to three times. Finally, the cell pellet was resuspended in a DMEM-F12 medium. Cell concentration and viability were evaluated by trypan blue staining.

We isolated viable singlet cells and subjected them to subsequent scRNA-seq, as previously reported[12]. Dissociated HCC cells were washed once in phosphate-buffered saline (PBS) with a centrifugation inflow tube. The cell pellet will be resuspended in a buffer of PBS with 2% fetal bovine serum (FBS) at a ratio of 50 μl per $1 \times 10^5$ cells. 7AAD (559925, BD, NJ, USA) dye was added at 1:10 and incubated in dark for a period of 20 min. The cells were finally washed once in FACS buffer and resuspended at a final concentration of $2 \times 10^6$/ml for subsequent cell sorting. FACS was carried out at the HKU Faculty Core Facility: 7AAD-positive cells (dead cells) and multiplet cells were gated out and not collected. Only 7AAD-negative (viable cells) singlet cells were sorted into the respective receiving tubes containing FACS buffer (PBS with 2% FBS), according to the concerned fluorochrome signal intensities. The collected cells were finally subjected to cell counting by trypan blue staining to evaluate the accurate cell concentration in order to facilitate downstream single-cell capture procedures.

**Single-cell capture, library preparation, and sequencing**. The viable singlet cell suspension was prepared to the desired range of concentration (100–2000 cells/μl), as recommended by the user protocol of 10× Genomics. It was then subjected to single-cell capture using the Chromium platform (10× Genomics, CA, USA). Chromium platform is a droplet-based system in which HCC single cells, gel beads with barcoded oligos, and reagents were mixed and captured as droplets in oil emulsion. Gene expression application was performed. Pooled and barcoded single cells in the form of droplets were then subjected to subsequent manipulation procedures, completing cell lysis, mRNA capture, reverse transcription, cDNA amplification, and Illumina sequencing library preparation (Illumina, San Diego, USA). Sequencing libraries of pooled single cells were obtained and sequenced by the Illumina Novaseq platform to provide adequate coverage.

**Preprocessing of scRNA-seq data**. We used Cell Ranger (version 3.0) from 10× Genomics to perform initial data demultiplexing, read alignment, UMI counting, and annotation on the raw read data. We tested for potential batch effect on experiment/sequencing by running principle component analysis (PCA) on the combined dataset, visual inspection of spatial stratification of non-malignant cells, and kBET algorithm (v0.99.5)[61]. Resulting gene-barcode matrices (one for each HCC case) were subjected to further quality control filtering, data preprocessing, scaling and normalization by Seurat package (v2.3)[62], with default parameters unless otherwise specified.

**Delineation of cellular and transcriptomic landscapes in HCCs**. We utilized various specific cell type markers for cell-type classification. We applied UMAP algorithm[13] to perform nonlinear dimensionality reduction. Single cells were stratified in two-dimension space, and the spatial separation of cells indicated their underlying similarities/differences in terms of their global transcriptomic landscape. To obtain the gene expression characteristics of different cell types, we performed an unsupervised graph-based Louvain clustering algorithm on the KNN graph, to group the single cells into different clusters with distinctive molecular signatures.

**Subclonal heterogeneity of tumor cells**. We examined the LCSC markers[12,63] and their combinatorial expression pattern and tested whether they marked any subclones of tumor cells. Given that RTKs are key components in transducing oncogenic signaling in cancers[64], the RTK family is a key vulnerability of HCC and currently the target of several systemic treatments using RTK inhibitors for advanced HCC. We further investigated the subclonal heterogeneity in terms of the expression landscape of LCSC markers and the RTK family.

**Copy number variation inference and lineage hierarchy of tumor cells**. We inferred the copy number variation (CNV) status, following the method by Tirosh et al.[20]. Copy number information was estimated by sorting the analyzed genes by their chromosomal location and applying a moving average to the relative expression values, with a sliding window of 100 genes within each chromosome. To infer copy number alterations, the copy number status of HCC tumor cells was tested against that of normal hepatocytes, taking the hepatocytes from the normal liver dataset of MacParland et al.[65] as reference. The inferred copy number alteration profiles of HCC tumor cells were used to derive lineage hierarchy in human HCCs.

**Immunosuppressive landscape in human HCC**. We examined the expression status of different complementary immune checkpoint molecules in T cells and APCs[66]. Both co-stimulatory and co-inhibitory immune checkpoints were evaluated for a comprehensive overview of the immunosuppressive landscape. We estimated the empirical significance of immune checkpoint interactions by evaluating the actual interaction score (the product of the mean of ligand expression in APCs and the mean of receptor expression in T cells) against the null distribution generated by permuting the cell type labels (but keeping the original cell type ratio) for 1000 times and calculated the P value as the proportion of permutated interaction scores that were greater than the actual interaction score. We considered both statistical significance and the actual interaction level to assess the importance of various immune checkpoints. With such, the most prominently expressed or the key immune checkpoint axis in the various complementary cell types were defined.

**Trajectory inference for tumor-infiltrating immune cell subpopulations**. The status of the tumor-infiltrating immune cell subpopulations in the TME is dynamic and they may differentiate into different cellular states that exert different biological functions, e.g., cancer fighting or cancer tolerant. We performed the trajectory analysis using pseudo-time inferencing algorithm Monocle 2[67] to reconstruct the cell differentiation trajectory of different tumor-infiltrating immune cells. It uses a machine-learning technique called reversed graph embedding to describe multiple fate decisions in a fully unsupervised manner and derives a principal tree on a population of single cells that reveals the progression of cell and reconstruct their trajectory as a cell progresses through the biological process under study. Different branches in the cell trajectory likely distinguished molecularly distinct cell subpopulations (denoted by different cellular states) within a certain cell type.

**Cell-cell interactome landscape between various tumor-residing cells**. We quantified the cell–cell interaction phenomenon using the established scoring algorithm[17] on 2557 curated ligand-receptor pairs[68]. We interrogated the interaction between immune cells (namely CD8 T, CD4 T, Treg, NK, and B cells) and APCs (TAM and tumor cells), with one of them contributing either ligand or receptor in the interaction process. To identify overall significant interactions that exist among multiple cell types, we performed a one-sided one-sample Wilcoxon signed-rank test to test whether the median interaction score across all cell types was greater than zero. We also applied the Storey method[69] to calculate the false discovery rate (FDR) that corrects for multiple testing. Interactions that having a score > 0.5 and FDR q < 0.05 were regarded as significant.

**T cell proliferation assay**. Splenic T cells from 5-to-7-week old C57/BL6N male mice were isolated with Mouse T Cell Enrichment Columns (R&D Systems, Minneapolis, USA) according to the manufacturer's instruction. Isolated T cells were labeled with 5 μM CellTrace Violet (CTV) (C34557, Invitrogen, CA, USA). Labeled T cells were cocultured with Hepa1–6 cells (WT, Nectin2-KO1, -Nectin2-KO2, -Nectin2-KO3) in a 1:1 ratio for 3 days (sgRNA sequences are listed in Supplementary Table 10). In the coculturing experiment of T and parental Hepa1–6 cells, 15 μg/mL anti-Nectin2 neutralizing antibody (MAB3869, R&D Systems, MN, USA) (Supplementary Table 11) was added.

**Mouse hydrodynamic tail–vein injection and orthotopic implantation models**. For hydrodynamic tail–vein injection (HDTVi), genome editing constructs in sterile saline constituted a total volume of 10% of the mouse body weight were injected into the lateral tail vein of C57BL/6N mice in 6–8 s. CRISPR–Cas9 vector system carrying sgRNAs targeting Trp53 and Nectin2 together with the Sleeping Beauty Transposon system overexpressing c-Myc vector was injected into the lateral tail vein of mice. HDTVi-induced tumors were harvested 5 weeks after HDTVi. For orthotopic implantation, $3 \times 10^6$ Hepa1–6–NTC and Hepa1–6–shNectin2 cells in matrigel were injected into the left lobes of the livers of C57BL/6N mice. Tumors were harvested 10 days after implantation. The sgRNA sequence for Nectin2 KO in HDTVi and the shRNA sequence for Nectin2 KD are listed in Supplementary Table 10.

**Animal studies**. Bodyweight was closely monitored. Euthanasia was performed if the percentage of body weight gain was greater than 10% due to tumor burden or weight loss was greater than 20%. Humane endpoint (HEP) scoring was taken daily to monitor symptoms of suffering according to the HEP scoring system established by The Centre for Comparative Medicine Research of The University of Hong Kong. If the total score was greater than four for three consecutive days or greater than six for any single day, euthanasia was performed. Euthanasia was performed by administration of Pentobarbitone intraperitoneally. Animal procedures within this study were approved by the Committee on the Use of Live Animals in Teaching and Research of the University of Hong Kong and adhered to the Animals (Control of Experiments) Ordinance of Hong Kong. The tumor size was represented by tumor size (mm³) = 0.5236 × length × width × height.

**Preparation of single-cell suspension from tumor tissues.** Single-cell suspensions from mouse tumors were prepared prior to flow cytometry analysis. Tumors harvested from HCC-bearing mice were cut into fine pieces and were suspended in DMEM/F12 medium. Totally, 30 μL of Liberase (2.5 mg/mL; 5401119001, Sigma Aldrich, St Louis, USA) and 60 μL of DNase I (10 mg/mL; 11284932001, Sigma Aldrich) were added. Tissues were dissociated with GentleMACS Dissociator (130-093-235, Miltenyi Biotech, Germany). Red blood cells were lysed with ACK lysing buffer. The filtered cell suspensions were washed and reconstituted in cell staining buffer for flow cytometry (BD, NJ, USA).

**Cell staining for flow cytometry.** Cells were blocked with anti-mouse CD16/32 antibody (101320, Biolegend, San Diego, USA) at room temperature for 10 min and incubated with fluorochrome-conjugated primary antibodies at 4 °C for 30 min. Primary antibodies and the dilutions are listed in Supplementary Table 11. CD4 T cells were $CD45^+CD3^+CD4^+$ cells and CD8 T cells are $CD45^+$ $CD8^+$ cells, respectively. T effector cells were $CD44^+CD62L^-$ T cells. Tregs are $CD45^+CD3^+$ $CD4^+CD25^+FoxP3^+$ T cells. Software FlowJo was used for data analysis. Gating strategies for tumor-infiltrating lymphocytes analysis are provided in Supplementary Fig. 22.

**IHC staining in human HCC samples.** IHC was performed on formalin-fixed, paraffin-embedded sections, as previously described[70–72], using anti-NECTIN2 (Sigma-Aldrich, HPA012759, 1:300), anti-TIGIT (Abcam, Ab243903, 1:200), anti-CD163 (Abcam, Ab189915, 1:500), and anti-LAIR1 (Sigma-Aldrich, HPA011155, 1:500) rabbit antibodies.

**IHC staining in mouse HCC samples.** Mouse HCC tissues fixed in 10% formalin and 75% ethanol were embedded in paraffin. Prior to IHC staining, paraffin sections were dewaxed with xylene and rinsed with ethanol. Antigen retrieval was achieved by boiling in 1 mM EDTA buffer (pH 7.8) for 15 min. Sections were blocked with 2× casein (SP-5020-250, Vector Laboratories, CA, USA) and stained with primary antibody (Supplementary Table 11) at 4 °C overnight. Sections were then stained with horseradish peroxidase-conjugated secondary antibody (Dako, Glostrup, Denmark) at room temperature for 30 min. Sections were developed using 3,3′-diaminobenzidine (DAB) (Sigma-Aldrich, St. Louis, USA) and counterstained with hematoxylin.

**Multicolor immunofluorescence staining.** We used Opal Polaris 7 Color Kit (NEL861001KT) according to the manufacturer's recommendation, with anti-CD163 (Abcam, Ab189915, 1:200) and anti-LAIR1 (Sigma-Aldrich, HPA011155, 1:500) rabbit antibodies.

**Cell proliferation and T cell coculture assay of THP-1 cells.** A stable *LAIR1* KD (sh*LAIR1*) clone was established in THP-1 cells (ATCC) by using the shRNA sequence (Supplementary Table 10) from Sigma-Aldrich. After establishing the sh*LAIR1* clone by lentiviral system[12], $2 \times 10^5$ live THP-1 cells were seeded per well in 24-well plate and MilliCell 0.4 μm-membrane insert (PSHT010R5, Sigma-Aldrich, MO, USA) with ~$2 \times 10^4$ PLC/PRF/5 cells were placed into the well of the 24-well plate for coculture of HCC cells and THP-1 cells in the presence of phorbol-12-myristate-13-acetate (PMA) (P1585, Sigma-Aldrich, Missouri, USA). After 24 h, the MilliCell 0.4 μm-membrane insert was replaced with another one which contained ~$2 \times 10^5$ human CD8 T cells in the presence of IL2 and anti-CD3/CD28 beads for coculture for five more days. The CD8 T cells were finally harvested and the anti-CD3/CD28 beads (11161D, Gibco, Thermo Fisher Scientific, Waltham, USA) were removed by MACSiMAG Separator (130-092-168, Miltenyi Biotec, Germany) according to the manufacturer's instruction before being washed once in PBS or RPMI1640 medium with 10% FBS. Staining of the CD8 T cells by CD44 and CD62L antibodies was done as described above. Flow cytometry was performed on a Novocyte Advanteon BVR flow cytometer machine in the core facility of the University of Hong Kong. For the study of the proliferation of THP-1 cells, 5000–10,000 cells were seeded per well on a 96-well plate in the presence of PMA. As THP-1 cells became adherent upon PMA treatment, the cells were trypsinized and counted on Day 6 after seeding the cells.

**Statistical analysis.** We performed statistical analysis using GraphPad Prism 6. A two-sided Student's *t* test was used to compare the means between two groups.

**Reporting summary.** Further information on research design is available in the Nature Research Reporting Summary linked to this article.

## Data availability
The sequence data that support the findings of this study have been deposited in the NCBI Sequence Read Archive under the primary accession code SRP318499. All other data are included in the article and its Supplementary Information files or available from the corresponding authors upon reasonable request. Source data are provided as a Source Data file. Source data are provided with this paper.

## Code availability
Custom R code for the CNV analyses included in the study can be downloaded from https://github.com/dwhho/scCNV.

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

## Acknowledgements

The study was supported by the Hong Kong Research Grants Council Theme-based Research Scheme (T12-704/16-R), Innovation and Technology Commission grant for State Key Laboratory of Liver Research, National Natural Science Foundation of China (81872222), and University Development Fund of The University of Hong Kong. I.O.L. Ng is Loke Yew Professor in Pathology.

## Author contributions

Study concept and design: D.W.H. and I.O.N.; Acquisition of data: D.W.H., Y.T., L.K.C., K.M.S., X.Z., X.Z., Y.T.C., J.M.L., J.W.C., C.C.W., and I.O.N.; Analysis and interpretation of data: D.W.H., Y.T., L.K.C., P.C.S., C.C.W., and I.O.N.; Acquisition of clinical samples: A.C.C., E.T.C., D.T.Y., N.H.C., I.L.L., T.T.C., and I.O.N.; Drafting of the paper: D.W.H., Y.T., and I.O.N. All authors reviewed and approved the final draft of the paper.

## Competing interests

The authors declare no competing interests.
