## [Peer Review File · Nature Communications]

REVIEWER COMMENTS

Reviewer #1 (Remarks to the Author):

In this study authors use single-cell RNA sequencing to investigate the immunological landscape in HCC in order to aid the development of novel immunomodulatory therapies. The study is well conceived with detailed analysis of tumour tissue which revealed the important role of the TIGIT-NECTIN2 axis which is then investigated in animal models. Whilst the data is interesting and well described there are several issues the authors need to address:

1. In the abstract and throughout the manuscript the authors state that the aim of the study is to delineate the immune landscape and tumour heterogeneity of human HCC however this should be restricted to the understanding of HBV+ HCC as all the samples were from HBV+ subjects.

2. How do the authors explain the relatively low interaction between PD1-PDL1/2 observed? Most studies suggest this to be a central mechanism of immune evasion. Multiple studies show high expression of PDL1 is high on HCC tumours and expression of PD1 high on TILs and in PBMC from HCC patients. Lack of efficacy of anti-PD1 or anti-PDL1 does not imply low expression as these have not been found to be biomarkers of response.

3. Authors suggest that the TIGIT-NECTIN2 pathway is important in HCC and confers a good IO target for HCC. Do the authors have any data from patients with chronic hepatitis B without HCC? Is this a viral evasion mechanism or a tumour evasion strategy? This could also be possibly answered by investigating the peri-tumour environment.

4. Following on from the question above, have the authors investigated the following in larger cohorts of HCC patients:

- a. Expression of TIGIT on PBMC/TIL from patients with HCC by FACS?
- b. Expression of NECTIN increased in tumour tissue by IHC or IF?

5. There are many checkpoint receptors and ligand and it is widely believed that there is a level of redundancy in these networks. For example if PD1 is knocked out or silenced other inhibitory checkpoints are upregulated. In the animal models used in the current study did the authors:

- a. Check whether other checkpoints were upregulated?
- b. Use neutralising antibodies rather than silencing or knock out models?

Minor comments:

1. The title should be more specific and give an insight into the main finding of the study.

2. Within the text and figure legends for Figure 2, the authors refer to Figure D and Figure E incorrectly as labelled in the figures which needs to be corrected.

3. In the text, the authors state 'Five weeks after HDTV_i, we found that the Nectin2 KO HCC tumors were significantly smaller in size as compared to Nectin2 WT HCC tumors (Figure 5b)'. However, the p-value for the tumour weight between WT and KO is not significant in figure 5b.

4. In the text, authors state 'Consistent with KO of Nectin2, knockdown (KD) of Nectin2 suppressed HCC growth and restored CD4 and CD8 T cell infiltration (Figure 5e-5g)'. Although the p-value for the difference in percentage of infiltrating CD4+ T-cells between NTC and shNectin2 is not indicated, there does not seem to be a difference between the two groups. Additionally, this statement is misleading when stating that there is an increase in CD8+ infiltrating T-cells as the p-value is not significant.

5. For Figure 5 C, D, F and G, please state the statistical analyses performed for the frequency of T cells and indicate the p-value on the figure or the figure legend. The authors need to be consistent in reporting p-values. If the comparison is not significant, this needs to be indicated.

6. In Figure 5 D, F and G, please indicate the percentage of PD1+ T-cells, TIGIT+ T-cells, CD4 T-cells and CD8-T cells in the FACS density plots

Reviewer #2 (Remarks to the Author):

In the paper by Ho et al., the authors performed a single-cell RNA sequencing of 9 HBV-related HCC samples using 10X Chromium platform and analyzed gene expression and cell-to-cell interactions with a special focus on the TIGIT-NECTIN2 axis. Overall, while the selection of patients' tissues is relatively homogenous and the cell filtering steps are rigorous, the results, data presentation and conclusions are not convincing. Indeed, the full picture of the microenvironment cell-to-cell interaction cannot be performed because endothelial cells are missing in the dataset. Importantly, no presented data can confirm an immunosuppressive role of LAIR1 in M2 macrophages nor that macrophages shape T-cell phenotype via TIGIT-NECTIN2 axis. NECTIN2 is highly expressed in tumor cells (higher than in macrophages). The functional data are very limited and not conclusive: they are focussed on tumor-T-cell interactions rather than macrophage-T-cell interactions which would be much more relevant. The mouse data do not show any significant findings and the results are largely overstated in the text. In summary, the study remains at a preliminary and largely descriptive level without arresting conclusions.

Major comments:

- NECTIN2 is highly expressed in tumor cells (higher than in macrophages) and all the functional experiments performed investigate the tumor-T-cell interaction rather than macrophage-T-cell interactions. There is no evidence that macrophage-T-cell interaction via TIGIT-NECTIN2 is relevant or more relevant than the tumor epithelial cell and T cell interaction.
- The association between LAIR1 and M2 markers is not enough to claim that LAIR1 plays a role in immunosuppression.
- Introduction: there are some wrong statements. The study by Ma et al. included 19 patients, 6 of 19 HCV.
- Data availability: "The data that support the findings of this study are available from the corresponding authors upon reasonable request." -> Data for Nature journal articles must be published at least in NCBI Trace Archive or NCBI Sequence Read Archive (SRA) or Gene Expression Omnibus (GEO). This is mandatory according to Nature reporting standards. Accession numbers must be provided in the paper.
- More clinical data should be reported in Supp Table 1 such as HBeAg status, HBV-DNA, HDV status, metabolic factors.
- Why are the reported statistics in Suppl. table 2 "preliminary"?
- Selection criteria for patients according to M&M "... (2) more than 1000 viable single cells obtained in each case." But patients #104 and #106 are listed with only 533 and 760 cells, resp., even before QC. This is somehow contradicting.
- Even if these criteria were applied after sorting, it results in not enough cells after sequencing since other patients were sequenced with >10.000 cells.
- The very different numbers of cells covered in patients (between 311 and 5.177 per patient after QC) suggest a bias of the results to the higher covered patients. For example, patients #104 and #106 represent only 2% and 3% of the whole data set, resp., while nearly 1/3 of all cells come from patient #114, and 1/4 from patient #095. Therefore, this data set with 55% cells originating only from 2 patients is highly unbalanced. This should be compensated by sequencing more cells of the under-represented patient's samples. The unbalanced data set is also reflected in the cell types (Supp. table 4). For example, there are 5 T-cells from patient #713 but 2.268 from patient #095. In fact, patients #095, #114, and #119 stand for 88% of all T-cells. In the end, it seems as some results from this study are drawn from 3 patients, while 5 are completely under-represented.
- The mouse data are not conclusive. Indeed, no difference is significant, and the results are largely overstated in the text.
- A Pearson correlation of -0.113 as shown on Fig. 1d / TCGA data does not confirm an inverse correlation. It may be significant (because of n=371), but not relevant. This part is overstated with the presented data.
- Comparing Fig 1b and Supp Fig 3 it seems that, except few clusters, the majority of the clusters are patient-related.
- It is not clear how the markers in Fig 2b have been selected. Are these the top hit?
- Fig 2d, it seems that very few tumors are LAIR1. How the cut-off value was chosen?
- Fig 2e, please show also the 3 channels separated. It seems that LAIR1 is expressed not only in TAMS.
- The tumor cell clustering according to LCSC markers is arbitrary and an unbiased clustering

should be preferred.

- The cell-state transition trajectory analysis lacks novelty. A deeper analysis should be performed.

Minor comments:

- Fig 1b, label all the cell types.

- Fig 1, present here also the unbiased clustering in the UMAP (Supp Fig 3)

- Figure 3, TAM trajectory analysis. The cluster color labeling in the pseudotime panel, should be the same of the violin plot.

- Page 7, line 7, there is an error in the figure citation (it is d instead of e).

Reviewer #3 (Remarks to the Author):

In the manuscript by Ho et al. (NCOMMS-20-33000-T) entitled "Single-cell RNA sequencing unravels the immunosuppressive landscape and tumor heterogeneity of hepatocellular carcinoma" the authors try to identify new mechanisms involved in immunosuppression in hepatocellular carcinoma (HCC) using single cell RNA sequencing. Using this technique, the authors obtained data supporting the relevance of tumor-associated macrophages in suppressing tumor T cell infiltration and have identified the TIGIT-NECTIN2 axis as a potential key element to facilitate an immunosuppressive environment in HCC. In addition, they found a great heterogeneity of tumor cells when comparing different HCCs. Although, in general, the results are novel and the work has been properly carried out, there are some weak points that need to be addressed according to the following comments:

Major points:

1-The analysis of human HCC samples using single-cell RNA sequencing only includes 8 samples of HCCs caused by B hepatitis virus from stages III-IV, and among them, only one corresponds to a woman. Therefore, it is unclear that the results derived from this study can be extrapolated to other HCC types and can be useful for both men and women. To evaluate it, the authors should determine, at least, if Nectin2 and TIGIT expression is deregulated in all HCC subtypes (not only in those caused by B hepatitis virus) using public databases of HCC patients in the different stages. It is true that the authors have also used two HCC mouse models where the effect of Nectin2 deletion is evaluated (Fig. 5b and 5e). However, data derived from these models are not always statistically significant. In particular, the decrease in tumor weight upon Nectin2 deletion (Fig. 5b) and the changes in the different subpopulations of infiltrated T cells (e.g. data from fig. 5c and 5g). Therefore, it is not fully demonstrated that Nectin2 deletion can restore the infiltration of T cells in HCC tumors and it needs to be done.

2-Based on the results derived from figure 5a, the authors claimed that Nectin2 suppresses proliferation of T cells. However, most of the results are not statistically significant. It is only found a significant increase in the number of CD4+ and CD8+ cells when Nectin2-KO2 hepatoma cells are co-cultured in the presence of T cells, but not in the case of Nectin2-KO1 cells. Therefore, additional experiments are required to determine if Nectin2 inhibits proliferation of T cells.

3-The authors have not included in the discussion, the information already available in the literature on the role of TIGIT in liver cancer, such as *Hepatology*. 2020 Jul 27. doi: 10.1002/hep.31466; *Nat Commun*. 2019 Jan 15;10(1):221. doi: 10.1038/s41467-018-08096-8 or even a previous manuscript from the authors cited in another section (*Gastroenterology*. 2020 Aug;159(2):609-623. doi: 10.1053/j.gastro.2020.03.074. (ref 16)). It would be important to discuss it to have a better idea of the novelty of the results as compared with published data.

Responses to Editor and Reviewers' comments

Reviewer #1 (Remarks to the Author):

In this study authors use single-cell RNA sequencing to investigate the immunological landscape in HCC in order to aid the development of novel immunomodulatory therapies. The study is well conceived with detailed analysis of tumour tissue which revealed the important role of the TIGIT-NECTIN2 axis which is then investigated in animal models. Whilst the data is interesting and well described there are several issues the authors need to address:

1. In the abstract and throughout the manuscript the authors state that the aim of the study is to delineate the immune landscape and tumour heterogeneity of human HCC however this should be restricted to the understanding of HBV+ HCC as all the samples were from HBV+ subjects.

Response: Thank you for pointing out this issue. We have revised our title and text to reflect our investigation of HBV-associated HCC.

2. How do the authors explain the relatively low interaction between PD1-PDL1/2 observed? Most studies suggest this to be a central mechanism of immune evasion. Multiple studies show high expression of PDL1 is high on HCC tumours and expression of PD1 high on TILs and in PBMC from HCC patients. Lack of efficacy of anti-PD1 or anti-PDL1 does not imply low expression as these have not been found to be biomarkers of response.

Response: We agree that lack of efficacy of PD-1 family of immunotherapy does not imply their low expression. In fact, we postulate that it is the result of expressional heterogeneity in different subset of HCC cases. It is believed that PD-1 expressed on the surfaces of T cells interacts with PD-L1/2 expressed on the surfaces of tumor cells to elicit T cell exhaustion consequence, which helps evading anti-tumor immunity. This is observed in multiple human cancers, including HCC. Early study by Gao et al. 2009 suggests that overexpression of PD-L1/2 in HCC was associated with poorer prognosis. However, the PD-L1 expression in tumor cells of HCC patients could vary. In the study by Calderaro et al. in 2016, PD-L1 expression in neoplastic cells with immunohistochemistry (IHC) was observed in only 17% of cases (with a mean of 5% positive cells) in 217 HCC tumors of varied etiologies. More recently, as demonstrated by Pinato et al. in 2019 using IHC with 5 different antibodies on 100 HCC cases, around 70% of cases were tested PD-L1 negative in the tumors with all the tested antibodies. Similarly, PD-L1 was tested positive in at most 22% and 19% of tumor-infiltrating immune cells and infiltrating immune cells of non-tumorous cirrhotic tissues, respectively. Therefore, we anticipate a non-negligible level of heterogeneity in PD-L1 expression in human HCC. This could explain the difference in observation between our study and reported literature. Besides, the HBV-positive background of our cases could likely impose another layer of difference, as existing literature analyses an admixed or primarily non-viral background. In Liao et al. 2019, using a cohort of 304 HCC patients (~85% being HBV-positive), the membrane expression of PD-L2 was observed in 19.1% of tumor samples and no obvious expression of PD-L1 was detected on tumor cell membranes. More importantly, since there is limited efficacy for single-agent or combined therapy of PD-1 family of immune checkpoint blockade, the underlying mechanism for the remaining proportion of HCC patients remains elusive. As reviewed by Li et al. 2020, both objective response rate (ORR) and disease control rate (DCR) were evaluated by pooled analysis in HBV-positive and non-viral HCC cases of multiple PD-1/PD-L1 inhibitor trials. While DCR is valid for both mono- and combined PD-1/PD-L1 inhibitor therapies and HBV-positive

cases achieved comparable ORR to those of non-viral cases, DCR was significantly lower in HBV-positive cases than the non-viral counterparts. Therefore, while it is not our intention to disprove the significance of PD-1 family of immune checkpoint blockade, we are suggesting the possibility of alternative TIGIT-NECTIN2 immune checkpoint axis in shaping the specific tumor immune microenvironment and achieving immune evasion in HBV-associated subset of human HCC. Our findings may help explaining the possible underlying mechanism in a specific portion of HBV-associated HCC cases, and would contribute to unravel a small but new area in the whole perspective.

Gao Q, et al. Overexpression of PD-L1 significantly associates with tumor aggressiveness and postoperative recurrence in human hepatocellular carcinoma. *Clin Cancer Res*. 2009 Feb 1;15(3):971-9. PMID: 19188168.

Calderaro J, et al. Programmed death ligand 1 expression in hepatocellular carcinoma: Relationship With clinical and pathological features. *Hepatology*. 2016 Dec;64(6):2038-2046. PMID: 27359084.

Pinato DJ, et al. Clinical implications of heterogeneity in PD-L1 immunohistochemical detection in hepatocellular carcinoma: the Blueprint-HCC study. *Br J Cancer*. 2019 May;120(11):1033-1036. PMID: PMC6738063.

Liao H, et al. Expression of programmed cell death-ligands in hepatocellular carcinoma: correlation with immune microenvironment and survival outcomes. *Front Oncol*. 2019 Sep 11;9:883. PMID: 31572677; PMID: PMC6749030.

Li B, et al. Anti-PD-1/PD-L1 Blockade Immunotherapy Employed in Treating Hepatitis B Virus Infection-Related Advanced Hepatocellular Carcinoma: A Literature Review. *Front Immunol*. 2020 May 28;11:1037. PMID: 32547550; PMID: PMC7270402.

3. Authors suggest that the TIGIT-NECTIN2 pathway is important in HCC and confers a good IO target for HCC. Do the authors have any data from patients with chronic hepatitis B without HCC? Is this a viral evasion mechanism or a tumour evasion strategy? This could also be possibly answered by investigating the peri-tumour environment.

Response: Thank you for your suggestion. We have analysed the expression of TIGIT and NECTIN2 using immunohistochemistry on a cohort of HCC (n=29) as well as a separate cohort of non-HCC HBV-associated cirrhotic liver (n=22). We examined the expression of TIGIT and NECTIN2 in lymphocytes and tumor cells, respectively. Interestingly, we were able to detect a statistically significant association between TIGIT and NECTIN2 expression in HCCs but not in the non-HCC cirrhotic livers. This suggests that the TIGIT-NECTIN2 axis is more likely to be a tumor evasion strategy, instead of the viral evasion one. These findings have been added in the Results section on p. 9 and in the new Supplementary Tables 6 and 7.

4. Following on from the question above, have the authors investigated the following in larger cohorts of HCC patients:

- a. Expression of TIGIT on PBMC/TIL from patients with HCC by FACS?
- b. Expression of NECTIN increased in tumour tissue by IHC or IF?

Response: We have performed immunohistochemistry examination of TIGIT and NECTIN2 on a cohort of HCC patients (n=29). We could detect substantial expression of TIGIT in tumor-infiltrating lymphocytes, which confirms our observation in scRNA-seq. The finding has been added in the Results on p. 9 and in the new Supplementary Table 6 and Supplementary Figure 9. Moreover, with IHC, we could also verify the significant

upregulation of NECTIN2 in HCC, as compared to the corresponding non-tumorous liver. The IHC finding has been added in the Results section on p. 9 and the new Supplementary Figure 10.

5. There are many checkpoint receptors and ligand and it is widely believed that there is a level of redundancy in these networks. For example, if PD1 is knocked out or silenced other inhibitory checkpoints are upregulated. In the animal models used in the current study did the authors:

- a. Check whether other checkpoints were upregulated?
- b. Use neutralising antibodies rather than silencing or knock out models?

Response: Based on the reviewer's suggestion, we have analysed multiple T cell inhibitory checkpoints (PD-1, TIGIT, LAG3, TIM3) in the Nectin2 WT and Nectin2 KO mouse HCC tumors. We found that there was no significant change in all the exhausted markers despite the number of CD4⁺ and CD8⁺ T cell infiltrates were significantly increased in the Nectin2 KO HCC tumors. We speculate that the expressions of exhaustion markers rely on the stage of the tumors. Our current hydrodynamic tail vein injection (HDTV_i) mouse model represents an advanced stage mouse HCC model. Many exhausted T cells at this stage might have undergone apoptosis, which could not be included in the flow analysis. The new data on the expression of these exhausted markers on the T cells have been included in the Supplementary Figure 11b of the revised manuscript.

Besides, based on the reviewer's suggestion, we have additionally performed our HCC:T cell co-culturing experiment with the Nectin2 neutralizing antibody. We observed that HCC cell-mediated suppression of T cell proliferation (both CD4⁺ and CD8⁺ T cells) could be restored by Nectin2 neutralizing antibody. These important data have been included in the Results on p.9 & 10, and Figure 5a of the revised manuscript.

Minor comments:

1. The title should be more specific and give an insight into the main finding of the study.

Response: Given that we studied HBV-associated HCC in the current study, we have revised the title as "Single-cell RNA sequencing unravels the immunosuppressive landscape and tumor heterogeneity of HBV-associated hepatocellular carcinoma". However, in view of the word limit for the title (maximum 15 words), we are not able to include more information.

2. Within the text and figure legends for Figure 2, the authors refer to Figure D and Figure E incorrectly as labelled in the figures which needs to be corrected.

Response: Thank you for pointing this out. We have revised the figure legend accordingly.

3. In the text, the authors state 'Five weeks after HDTV_i, we found that the Nectin2 KO HCC tumors were significantly smaller in size as compared to Nectin2 WT HCC tumors (Figure 5b)'. However, the p-value for the tumour weight between WT and KO is not significant in figure 5b.

Response: We have repeated the experiment with increased sample size. In the revised Figure 5c, we can demonstrate statistically significant reduction in tumor weight upon Nectin2 KO, as compared to WT controls.

4. In the text, authors state 'Consistent with KO of Nectin2, knockdown (KD) of Nectin2 suppressed HCC growth and restored CD4 and CD8 T cell infiltration (Figure 5e-5g)'. Although the p-value for the difference in percentage of infiltrating CD4⁺ T-cells between

NTC and shNectin2 is not indicated, there does not seem to be a difference between the two groups. Additionally, this statement is misleading when stating that there is an increase in CD8⁺ infiltrating T-cells as the p-value is not significant.

Response: We have repeated the Nectin2 KO and KD *in vivo* experiments. In addition to flow cytometry analysis, we also performed IHC staining with CD4 and CD8 antibodies to confirm our results. Both flow cytometry and IHC studies demonstrated that the numbers of CD4⁺ and CD8⁺ T cell infiltrates were increased in the Nectin2 KO HCC tissues with statistical significance ($P < 0.05$). The new data are now included in Figure 5e-5h of the revised manuscript. Data for Nectin2 KD experiment could be found in the Supplementary Figure 11c-11e. Statistical significance could also be achieved.

5. For Figure 5 C, D, F and G, please state the statistical analyses performed for the frequency of T cells and indicate the p-value on the figure or the figure legend. The authors need to be consistent in reporting p-values. If the comparison is not significant, this needs to be indicated.

Response: In the revised manuscript, we have indicated the statistical analyses in the figure legend. Comparisons between experimental groups were clearly indicated with lines. Student's t test was employed in all comparisons. The precise P values were indicated by asterisks as follows: * $P < 0.05$, ** $P < 0.01$, *** $P < 0.001$, **** $P < 0.0001$. If the comparison was not statistically significant, we have indicated the P values as NS (no significance).

6. In Figure 5 D, F and G, please indicate the percentage of PD1⁺ T-cells, TIGIT⁺ T-cells, CD4 T-cells and CD8-T cells in the FACs density plots.

Response: We have indicated the cell percentages in the density plots of the revised Figure 5 and Supplementary Figure 11.

Reviewer #2 (Remarks to the Author):

In the paper by Ho et al., the authors performed a single-cell RNA sequencing of 9 HBV-related HCC samples using 10X Chromium platform and analyzed gene expression and cell-to-cell interactions with a special focus on the TIGIT-NECTIN2 axis. Overall, while the selection of patients' tissues is relatively homogenous and the cell filtering steps are rigorous, the results, data presentation and conclusions are not convincing. Indeed, the full picture of the microenvironment cell-to-cell interaction cannot be performed because endothelial cells are missing in the dataset. Importantly, no presented data can confirm an immunosuppressive role of LAIR1 in M2 macrophages nor that macrophages shape T-cell phenotype via TIGIT-NECTIN2 axis. NECTIN2 is highly expressed in tumor cells (higher than in macrophages). The functional data are very limited and not conclusive: they are focussed on tumor-T-cell interactions rather than macrophage-T-cell interactions which would be much more relevant. The mouse data do not show any significant findings and the results are largely overstated in the text. In summary, the study remains at a preliminary and largely descriptive level without arresting conclusions.

Major comments:

1. NECTIN2 is highly expressed in tumor cells (higher than in macrophages) and all the functional experiments performed investigate the tumor-T-cell interaction rather than macrophage-T-cell interactions. There is no evidence that macrophage-T-cell interaction via TIGIT-NECTIN2 is relevant or more relevant than the tumor epithelial cell and T cell interaction.

Response: As indicated in Figure 4b, NECTIN2 is highly expressed in tumor cells, which is even higher than in macrophages. Therefore, we believe that the immunosuppressive effect of TIGIT-NECTIN2 axis is more prominently exerted via tumor-T cell interaction, but we do not rule out the possibility of macrophage-T cell interaction.

2. The association between LAIR1 and M2 markers is not enough to claim that LAIR1 plays a role in immunosuppression.

Response: We have performed additional immunohistochemistry examination of LAIR1 and CD163 (M2 macrophage marker) expression on a cohort of patients' HCCs. We examined the co-expression of LAIR1 and CD163. With this independent and expanded sample cohort, we could detect a statistically significant association between LAIR1 and CD163 expression in HCCs. The result additionally confirms our hypothesis of enriched LAIR1 expression in M2 macrophages. The finding has been added in the Results on p.7, Supplementary Table 5 and Supplementary Figure 5. Since M2 macrophages are known to be cancer-promoting, our findings suggest that the effect may likely be exerted via LAIR1 expression. Moreover, we have established stable LAIR1 knockdown (shLAIR1) macrophages using THP-1 cells. Upon shLAIR1, THP-1 cells demonstrate reduced proliferation. By co-culturing them with CD8 T cells, we also identified upregulated T cell activation, as exemplified by increased proportion of CD44+CD62L- effector T cells (Supplementary Figure 6). We agree that our data in this study may not be enough to support this claim and accordingly we have tuned it down in the manuscript.

3. Introduction: there are some wrong statements. The study by Ma et al. included 19 patients, 6 of 19 HCV.

Response: In the study by Ma et al. 2019, they collectively studied a cohort of 19 primary liver cancer patients (9 HCC and 10 intrahepatic cholangiocarcinoma [iCCA]). Since our current study focusses on HCC, we only refer to the 9 HCC cases (with majority of them [6 out of 9] being HCV infected) in the comparison.

Ma L, Hernandez MO, Zhao Y, et al. Tumor Cell Biodiversity Drives Microenvironmental Reprogramming in Liver Cancer. *Cancer Cell*. 2019 Oct 14;36(4):418-430.e6. PMID: 31588021; PMCID: PMC6801104.

4. Data availability: "The data that support the findings of this study are available from the corresponding authors upon reasonable request." -> Data for Nature journal articles must be published at least in NCBI Trace Archive or NCBI Sequence Read Archive (SRA) or Gene Expression Omnibus (GEO). This is mandatory according to Nature reporting standards. Accession numbers must be provided in the paper.

Response: Thank you for your pointing it out. We have uploaded the data online and it is publicly available.

5. More clinical data should be reported in Supp Table 1 such as HBeAg status, HBV-DNA, HDV status, metabolic factors.

Response: We have incorporated the additional information into the revised Supplementary Table 1.

6. Why are the reported statistics in Suppl. table 2 "preliminary"?

Response: Thank you for the comment. The information reported in Supp Table 2 refers to as “preliminary statistics” because they are the initial cell count, read count and percentages that calculated based on raw data. We have removed the term “preliminary” and revised the title as “Supplementary Table 2. Statistics of the scRNA-seq dataset.”.

7. Selection criteria for patients according to M&M "... (2) more than 1000 viable single cells obtained in each case." But patients #104 and #106 are listed with only 533 and 760 cells, resp., even before QC. This is somehow contradicting.

Response: The criterion of having more than 1000 viable single cells refers to the number of cells in the dissociated tumor cell suspension, before the single-cell capture using Chromium platform and the actual scRNA-seq.

8. Even if these criteria were applied after sorting, it results in not enough cells after sequencing since other patients were sequenced with >10,000 cells.

Response: Due to difference in cell viability and available cell count in each case, there is variation in the actual sequenced cell counts in our sample cohort. We recognize this limitation and have clearly stated the detailed cell counts in our dataset in Supp Table 2. This is intended to faithfully reflect the intrinsic nature of our dataset to the readers. Despite this limitation in the variation of the viable cell numbers, our findings have confirmatory support with multifaceted examination in patients’ HCC and also wet-lab experiments *in vitro* and *in vivo*.

9. The very different numbers of cells covered in patients (between 311 and 5.177 per patient after QC) suggest a bias of the results to the higher covered patients. For example, patients #104 and #106 represent only 2% and 3% of the whole data set, resp., while nearly 1/3 of all cells come from patient #114, and 1/4 from patient #095. Therefore, this data set with 55% cells originating only from 2 patients is highly unbalanced. This should be compensated by sequencing more cells of the under-represented patient's samples. The unbalanced data set is also reflected in the cell types (Supp. table 4). For example, there are 5 T-cells from patient #713 but 2.268 from patient #095. In fact, patients #095, #114, and #119 stand for 88% of all T-cells. In the end, it seems as some results from this study are drawn from 3 patients, while 5 are completely under-represented.

Response: The variable sequenced cell count in each case reflects the difference in cell viability and the available amount of tissue. Resected tumor tissues from HCC patients are our precious samples and we have already used up the available tissue sample for those selected HCC cases. For this, we cannot sequence more cells from those cases. However, we truly admit this limitation and have clearly stated the different sequenced cell counts and the cell type compositions. Despite this limitation in the variation of the viable cell numbers, our findings have confirmatory support with multifaceted examination in patients’ HCC and also wet-lab experiments *in vitro* and *in vivo*.

10. The mouse data are not conclusive. Indeed, no difference is significant, and the results are largely overstated in the text.

Response: We have repeated all animal experiments (revised Figure 5 and Supplementary Figure 11). Furthermore, we also repeated all T cell: HCC co-culturing experiments with additional Nectin2 KO clones and Nectin2 neutralizing antibody (revised Figure 5). In addition to flow cytometry analysis, we also included IHC staining to confirm our data on increase of CD4⁺ and CD8⁺ T cell infiltrates in Nectin2 KO and KD HCC tumors as

compared to their control tumors. We confirmed that Nectin2 depletion significantly reduced tumor size and increased CD4⁺ and CD8⁺ T cells. The P values were indicated in all experiments. Furthermore, depletion of Nectin2 using neutralizing antibody as well as KO approach in HCC cells also restored T cell proliferation in HCC:T cell co-culturing experiments. Statistical significance could be achieved in all the above experiments and indicated in the Figure 5 and Supplementary Figure 11 of the revised manuscript.

11. A Pearson correlation of -0.113 as shown on Fig. 1d / TCGA data does not confirm an inverse correlation. It may be significant (because of n=371), but not relevant. This part is overstated with the presented data.

Response: Thank you for the comment. We have revised our statement in the manuscript to only suggest this possibility of inverse correlation.

12. Comparing Fig 1b and Supp Fig 3 it seems that, except few clusters, the majority of the clusters are patient-related.

Response: The results indeed suggest that the majority of cell clusters are patient-related i.e. HCC tumor cells mainly group together according to their case identities, with very few cells admixed into clusters from other cases. This indicates a high degree of inter-tumoral heterogeneity. This also indicates HCC tumor cells from the same patient shared a higher degree of similarity than those from different patients.

13. It is not clear how the markers in Fig 2b have been selected. Are these the top hit?

Response: Yes, they are the top hit markers for the TAM cell clusters.

14. Fig 2d, it seems that very few tumors are LAIR1. How the cut-off value was chosen?

Response: The analysis was done on the TCGA LIHC dataset via cBioPortal. We used the default z-score threshold of 2 to define the relative level of LAIR1 expression. We have revised the text to incorporate this information.

15. Fig 2e, please show also the 3 channels separated. It seems that LAIR1 is expressed not only in TAMs.

Response: We have revised the Figure 2e, with both the images on combined and separated channels. Data suggests LAIR1 and CD163 expressions are largely overlapping. In addition, we have performed IHC for LAIR1 and CD163 on a cohort of patients' HCC. A representative figure showing overlapping staining is added (Supplementary Figure 5).

16. The tumor cell clustering according to LCSC markers is arbitrary and an unbiased clustering should be preferred.

Response: We have performed the unbiased clustering and the result is shown in Figure 1c.

17. The cell-state transition trajectory analysis lacks novelty. A deeper analysis should be performed.

Response: The cell trajectory analysis was performed to evaluate the transition of cellular status of different immune cells in general. It is reflecting the *bona fide* immunosuppressive landscape that involves the endeavor of different immune cells to result in immune invasion allowing HCC development. The major novelty of this part is the suggestion of the involvement of exhausted TIGIT-expressing CD8 T cells and immunosuppressive LAIR1-

expressing M2 macrophages. The cell trajectory analysis complements the investigation in other parts, with consistent suggestion on the involvement of these 2 subsets of immune cells in HCC.

Minor comments:

1. Fig 1b, label all the cell types.

Response: Cell type labels have been added in Figure 1b.

2. Fig 1, present here also the unbiased clustering in the UMAP (Supp Fig 3)

Response: We have incorporated the unbiased clustering (original Supplementary Figure 3a) as Figure 1c.

3. Figure 3, TAM trajectory analysis. The cluster color labeling in the pseudotime panel, should be the same of the violin plot.

Response: We have modified the colors of the violin plots in Figure 3.

4. Page 7, line 7, there is an error in the figure citation (it is d instead of e).

Response: Thank you for pointing out this typo. We have amended the text accordingly.

Reviewer #3 (Remarks to the Author):

In the manuscript by Ho et al. (NCOMMS-20-33000-T) entitled “Single-cell RNA sequencing unravels the immunosuppressive landscape and tumor heterogeneity of hepatocellular carcinoma” the authors try to identify new mechanisms involved in immunosuppression in hepatocellular carcinoma (HCC) using single cell RNA sequencing. Using this technique, the authors obtained data supporting the relevance of tumor-associated macrophages in suppressing tumor T cell infiltration and have identified the TIGIT-NECTIN2 axis as a potential key element to facilitate an immunosuppressive environment in HCC. In addition, they found a great heterogeneity of tumor cells when comparing different HCCs. Although, in general, the results are novel and the work has been properly carried out, there are some weak points that need to be addressed according to the following comments:

Major points:

1. The analysis of human HCC samples using single-cell RNA sequencing only includes 8 samples of HCCs caused by B hepatitis virus from stages III-IV, and among them, only one corresponds to a woman. Therefore, it is unclear that the results derived from this study can be extrapolated to other HCC types and can be useful for both men and women. To evaluate it, the authors should determine, at least, if Nectin2 and TIGIT expression is deregulated in all HCC subtypes (not only in those caused by B hepatitis virus) using public databases of HCC patients in the different stages. It is true that the authors have also used two HCC mouse models where the effect of Nectin2 deletion is evaluated (Fig. 5b and 5e). However, data derived from these models are not always statistically significant. In particular, the decrease in tumor weight upon Nectin2 deletion (Fig. 5b) and the changes in the different subpopulations of infiltrated T cells (e.g. data from fig. 5c and 5g). Therefore, it is not fully demonstrated that Nectin2 deletion can restore the infiltration of T cells in HCC tumors and it needs to be done.

Response: HCC is a male-predominant malignancy. In the recent publication from our center (Chan MY et al. Transl Gastroenterol Hepatol 2019 Jul 11;4:52), the overall male:female

ratio of the HCC patients receiving curative hepatectomy was 4.07. Our cases in this current study were all HBV-associated and the findings should be applicable to at least HBV-associated HCC group. As suggested by the other reviewer, we have also amended the title of this manuscript to HBV-associated hepatocellular carcinoma.

We have repeated the Nectin2 KO and KD *in vivo* experiments. In addition to flow cytometry analysis, we have also performed IHC staining with CD4 and CD8 antibodies to confirm our results. Both flow cytometry and IHC studies consistently demonstrated that the numbers of CD4⁺ and CD8⁺ T cell infiltrates were increased in the Nectin2 KO HCC tissues with statistical significance ($P < 0.05$). The new data are now included in the Figure 5e-5h of the revised manuscript. Data for Nectin2 KD experiment could be found in Supplementary Figure 11c-11e. Statistical significance could also be achieved.

2. Based on the results derived from figure 5a, the authors claimed that Nectin2 suppresses proliferation of T cells. However, most of the results are not statistically significant. It is only found a significant increase in the number of CD4⁺ and CD8⁺ cells when Nectin2-KO2 hepatoma cells are co-cultured in the presence of T cells, but not in the case of Nectin2-KO1 cells. Therefore, additional experiments are required to determine if Nectin2 inhibits proliferation of T cells.

Response: We have repeated our co-culturing experiments with additional Hepa1-6-Nectin2 KO clones. All 3 different Nectin2 KO clones significantly restored T cell proliferation as compared to WT. We have included the analyses in the revised manuscript (Figure 5b). In addition, to further substantiate our observation and as suggested by other reviewers, we have further confirmed our result by using Nectin2 neutralizing antibody. We found that Nectin2 neutralizing antibody, as similar to the Nectin2 KO in HCC cells, could significantly restore T cell proliferation (Figure 5a).

3. The authors have not included in the discussion, the information already available in the literature on the role of TIGIT in liver cancer, such as Hepatology. 2020 Jul 27. doi: 10.1002/hep.31466; Nat Commun. 2019 Jan 15;10(1):221. doi: 10.1038/s41467-018-08096-8 or even a previous manuscript from the authors cited in another section (Gastroenterology. 2020 Aug;159(2):609-623. doi: 10.1053/j.gastro.2020.03.074. (ref 16)). It would be important to discuss it to have a better idea of the novelty of the results as compared with published data.

Response: Thank you for the comment. We have included those studies in the discussion and indicated the reported roles of TIGIT in liver cancer. We have also highlighted the novelty and advancement in knowledge that provided by our current study.

REVIEWERS' COMMENTS

Reviewer #1 (Remarks to the Author):

The authors have resolved all my concerns and the additional experiments significantly increase confidence in the findings and the novelty of this work.

In particular, the expansion of the cohorts, the IHC analysis of cirrhotic vs HCC tissues and the use of the NECTIN neutralizing antibody improve the quality of the study.

Reviewer #2 (Remarks to the Author):

The authors addressed the key points and the manuscript has substantially improved.

Reviewer #3 (Remarks to the Author):

The revised version of the manuscript by Ho et al. (NCOMMS-20-33000-A) has been improved according to the comments to authors. The title has been changed as their studies only applied to HBV-associated hepatocellular carcinoma, but not to all HCC subtypes. Therefore, the new title refers only to HBV-associated HCC. In addition, new experiments and text modifications have been performed to solve the weak points of the study.